# p16$^{Ink4a}$ and p21$^{Cip1/Waf1}$ promote tumour growth by enhancing myeloid-derived suppressor cells chemotaxis

Atsushi Okuma [1,2], Aki Hanyu[2], Sugiko Watanabe [1] & Eiji Hara[1,2]

p16$^{Ink4a}$ and p21$^{Cip1/Waf1}$ act as tumour suppressors through induction of cellular senescence. However, senescence-independent roles of these CDK inhibitors are not well understood. Here, we report an unexpected function of p16$^{Ink4}$ and p21$^{Cip1/Waf1}$, namely, tumour promotion through chemotaxis. In monocytic myeloid-derived suppressor cells (Mo-MDSCs), p16$^{Ink4}$ and p21$^{Cip1/Waf1}$ are highly expressed and stimulate CX3CR1 chemokine receptor expression by preventing CDK-mediated phosphorylation and inactivation of SMAD3. Thus, deletion of p16$^{Ink4}$ and p21$^{Cip1/Waf1}$ reduces CX3CR1 expression, thereby inhibiting Mo-MDSC accumulation in tumours expressing CX3CL1 and suppressing the tumour progression in mice. Notably, blockade of the CX3CL1/CX3CR1 axis suppresses tumour growth, whereas inactivation of CDKs elicits the opposite effect. These findings reveal an unexpected function of p16$^{Ink4a}$ and p21$^{Waf1/Cip1}$ and indicate that regulation of Mo-MDSCs chemotaxis is a valuable potential strategy for control of tumour development.

[1] Department of Molecular Microbiology, Research Institute for Microbial Diseases, Osaka University, 3-1 Yamadaoka, Suita, Osaka 565-0871, Japan. [2] The Cancer Institute, Japanese Foundation for Cancer Research, Koto-ku, Tokyo 135-8550, Japan. Correspondence and requests for materials should be addressed to A.O. (email: aokuma@biken.osaka-u.ac.jp) or to E.H. (email: ehara@biken.osaka-u.ac.jp)

The $p16^{Ink4a}$ and $p21^{Waf1/Cip1}$ genes, which encode cyclin-dependent kinase (CDK) inhibitors, are upregulated in cultured mammalian primary cells upon detection of various potentially oncogenic stimuli[1,2]. This unique feature of $p16^{Ink4a}$ and $p21^{Waf1/Cip1}$, together with their ability to induce irreversible cell cycle arrest (termed cellular senescence), suggests that these genes act as a safeguard against neoplasia[3–5]. Indeed, mice lacking $p16^{Ink4a}$ and/or $p21^{Waf1/Cip1}$ exhibit early onset of cancer[6–9], illustrating the importance of $p16^{Ink4a}$ and $p21^{Waf1/Cip1}$ in tumour suppression in vivo. To observe the physiological roles of $p16^{Ink4a}$ and $p21^{Waf1/Cip1}$ during tumour formation, we previously generated transgenic mice lines expressing firefly luciferase under the control of the $p16^{Ink4a}$ or $p21^{Waf1/Cip1}$ gene promoters; these were termed p16-luc or p21-luc mice, respectively, and we revealed the timing, and hence, the likely roles of $p16^{Ink4a}$ and/or $p21^{Waf1/Cip1}$ expression in de novo tumorigenesis in vivo[10,11].

Similar to our work, Burd et al. reported the generation of another strain of $p16^{Ink4a}$ reporter mice ($p16^{Luc}$ mice), in which the $p16^{Ink4a}$ coding sequence was replaced with cDNA encoding firefly luciferase[12]. Notably, in addition to ageing and de novo tumorigenesis, $p16^{Ink4a}$ expression was strikingly induced in the stroma of developing neoplasia. Lethal irradiation coupled with bone marrow (BM) transplantation from syngeneic $p16^{Luc}$ mice indicated the presence of $p16^{Ink4a}$-expressing BM-derived cells in the tumour microenvironment[12]. However, it is not known which cell types express $p16^{Ink4a}$ in the stroma of developing neoplasia. Moreover, the biological role of $p16^{Ink4a}$ expression in these cancer stromal cells remains unclear.

In this study, we found that both $p16^{Ink4a}$ and $p21^{Waf1/Cip1}$ are highly expressed in myeloid-derived suppressor cells (MDSCs) without any noticeable senescence-associated phenotypes. MDSCs are known to negatively regulate immune responses and facilitate tumour progression[13]. The systemic expansion of MDSCs in BM and the MDSC migration to peripheral are the hallmarks of tumour-bearing condition[14]. In mice, MDSCs are defined as Gr-1$^+$CD11b$^+$ cells and categorised into 2 subtypes: polymorphonuclear (PMN-) and monocytic (Mo-) MDSCs[15,16]. Mo-MDSCs are a CD11b$^+$Ly6C$^{high}$Ly6G$^-$ subset that is highly immunosuppressive and exerts its effects in both antigen-specific and antigen-non-specific manners[15,16]. In contrast, CD11b$^+$Ly6C$^{int}$Ly6G$^+$ PMN-MDSCs are moderately immunosuppressive and act via antigen-specific mechanisms[15,16]. Mo- and PMN-MDSCs differ markedly in terms of the molecular mechanisms involved in their migration/chemotaxis. Mo-MDSCs express C-C chemokine receptor type (CCR)2, CCR5 and CX3C chemokine receptor (CX3CR)1[17,18], whereas PMN-MDSCs express CXCR2[19]. The accumulation of intratumoural MDSCs dependent on these chemokine receptors is well correlated with tumour growth[17,19].

In the present study, we reveal that $p16^{Ink4a}$ and $p21^{Waf1/Cip1}$ upregulate CX3CR1 expression by preventing CDK-mediated phosphorylation and inactivation of SMAD3 in Mo-MDSCs. Thus, deletion of $p16^{Ink4a}$ and $p21^{Waf1/Cip1}$ in mice results in a substantial decrease in infiltration of Mo-MDSCs into tumours and causes slower growth of tumour allografts. Conversely, inactivation of CDKs by chemical inhibitors increases the expression of CX3CR1 in Mo-MDSCs, resulting in accumulation of Mo-MDSCs in tumours and consequent acceleration of tumour growth in allograft mouse models. These results uncover a novel function of $p16^{Ink4a}$ and $p21^{Waf1/Cip1}$ in MDSC chemotaxis, and provide valuable new insight into how to bypass this undesirable side effect of CDK inhibitors.

## Results

### p16 and p21 are expressed in MDSCs in tumour-bearing mice.
We previously performed in vivo imaging of $p16^{Ink4a}$ or $p21^{Cip1/}$ $^{Waf1}$ expression in mice and elucidated the dynamics of their expression during the development of skin cancer, using p16-luc or p21-luc mice[9–11]. This approach, together with the analysis of $p16^{Ink4a}$ and/or $p21^{Cip1/Waf1}$-knockout mice, suggested that $p16^{Ink4a}$ and $p21^{Cip1/Waf1}$ are likely to be expressed in pre-malignant benign tumour cells, thereby having a protective role against malignant transformation. However, we were unable to exclude the possibility of $p16^{Ink4a}$ and $p21^{Cip1/Waf1}$ expression in non-tumour cells within the tumour microenvironment in these experimental settings. Thus, to ascertain this possibility, we inoculated p16- and p21-luc mice with syngeneic cancer cell lines, such as Lewis lung carcinoma (LLC) or spindle cell tumour (SCT) cells, neither of which expresses luciferase. Notably, inoculation of these cancer cell lines induced significant levels of luciferase activity in the area of tumour formation, but not in contralateral Matrigel-injected areas, in both p16-luc and p21-luc mice (Fig. 1a, b). This is consistent with a previous report describing the presence of BM-derived $p16^{Ink4a}$-expressing stroma cells in the tumour microenvironment[12]. Collectively, these results lead us to an idea that the expression of $p16^{Ink4a}$ and/or $p21^{Cip1/Waf1}$ may have important roles in establishing the tumour microenvironment during cancer development.

To explore this possibility further, we next attempted to identify the cell types expressing $p16^{Ink4a}$ and/or $p21^{Cip1/Waf1}$ in tumour stroma. In addition to cells in the tumour stroma, spleen cells in tumour-bearing (TB) mice expressed both $p16^{Ink4a}$ and $p21^{Cip1/Waf1}$ and were CD11b$^+$ and Gr-1$^+$, as indicated by immunohistochemistry (Fig. 1c–f). Peripheral CD11b$^+$Gr-1$^+$ cells are known to be MDSCs[20] and MDSCs are often induced by the presence of tumours in mice and humans[21,22]. Accordingly, PMN-MDSCs and Mo-MDSCs were purified from the spleens of tumour-bearing mice, and $p16^{Ink4a}$ and $p21^{Cip1/Waf1}$ mRNA levels were examined by quantitative real-time reverse transcription (qRT-) PCR (Fig. 1g, h). Interestingly, although $p16^{Ink4a}$ was expressed in both PMN-MDSCs and Mo-MDSCs, $p21^{Cip1/Waf1}$ was only expressed in Mo-MDSCs.

As $p16^{Ink4a}$ and $p21^{Cip1/Waf1}$ CDK inhibitors have established roles in cellular senescence, we tested if $p16^{Ink4a}$- and/or $p21^{Cip1/Waf1}$-expressing MDSCs exhibit senescence-like phenotypes. Consistent with a previous report[23], BM Mo-MDSCs are proliferative and the percentage of Mo-MDSCs in the S phase increases in mice lacking both $p16^{Ink4a}$ and $p21^{Cip1/Waf1}$ (p16/p21-DKO mice), compared to in wild-type (WT) mice (Supplementary Fig. 1a). On the other hand, in either splenic or intratumoural MDSCs, there is no difference in cell cycle phase distribution between MDSCs from WT mice and those from p16/p21-DKO mice (Supplementary Fig. 1a). Notably, although proliferation of MDSCs isolated from spleen was rarely detected by a 5-ethynyl-2′-deoxyuridine (EdU) incorporation assay in vivo (Supplementary Fig. 1b), a carboxyfluoroscein diacetate succinimidyl ester (CFSE) dilution analysis indicated that a substantial amount of these MDSCs (Mo-MDSCs >20%, PMN-MDSCs >60%) resumed proliferation upon stimulation with GM-CSF in vitro (Supplementary Fig. 1c). Moreover, other senescence-associated phenotypic characteristics, such as accumulation of γH2AX foci and 53BP1 foci (signs of DNA damage), reduction of lamin B1 expression[24], and induction of IL-6 expression[25], were not observed in these MDSCs (Supplementary Fig. 1d–g). These results, together with the observations that these MDSCs were resistant to ABT-263, a senolytic drug that specifically kills senescent cells[26], in both in vitro and in vivo (Supplementary Fig. 1h, i), indicate that these MDSCs are very unlikely to be in a state of cellular senescence despite their high expression of $p16^{Ink4a}$ and $p21^{Cip1/Waf1}$. These findings then raise questions about the roles of $p16^{Ink4a}$ and $p21^{Cip1/Waf1}$ expression in MDSCs.

**p16 and p21 in Mo-MDSCs promote tumorigenesis in vivo.** MDSCs have been reported to exert immunosuppressive effects and promote tumour development[14]. To verify the tumour-promoting effect of MDSCs expressing p16[Ink4a] and p21[Cip1/Waf1], WT and p16/p21-DKO mice were subcutaneously inoculated with orthotopic SCT cells. Surprisingly, SCT cells grew more slowly in p16/p21-DKO mice than in sex- and age-matched WT mice (Fig. 2a, b; Supplementary Fig. 2), suggesting that p16[Ink4a] and p21[Cip1/Waf1] have a tumour-promoting role, at least in the present experimental setting. The primary function of tumour-derived MDSCs is reportedly immunosuppression, especially inhibition of T cell activation[14]. We therefore investigated whether p16[Ink4a] and p21[Cip1/Waf1] are involved in immunosuppressive activity of MDSCs by assessing the activation of T cells co-cultured with MDSCs from WT or p16/p21-DKO mice. Purified PMN- and Mo-MDSCs were added to OVA-pulsed DC2.4 cells, a dendritic cell line[27], and to RF33.70 cells, an OVA (SIINFEKL)-specific T-T hybridoma[28]. T cell activation was evaluated by measuring interleukin (IL)-2 production by RF33.70 cells[29]. Unexpectedly, however, there was no significant difference in immunosuppressive capacity between WT and p16/p21-DKO

MDSCs (Supplementary Fig. 3a). Similar results were also obtained by performing a co-culture assay of splenic Mo-MDSCs or intratumoural Mo-MDSCs with purified splenic T cells stimulated with anti-CD3/CD28 antibody-coated beads (Supplementary Fig. 3b, c). In agreement with these observations, expression of the MDSC-derived immunosuppressive factors arginase (Arg)1[30] and nitric oxide synthase (Nos)2[31] was not downregulated in Mo-MDSCs lacking p16[Ink4a] and p21[Cip1/Waf1] (Supplementary Fig. 3d). The expression levels of Il1rn, which encodes IL-1RA and promotes tumour growth by inhibiting an inflammatory cytokine IL-1α[32], and Cd274, which encodes programmed death ligand (PD-L)1 and kills T cells[33], were similar between WT MDSCs and p16/p21-DKO MDSCs (Supplementary Fig. 3d). Therefore, it is unlikely that the expression of p16[Ink4a] and p21[Cip1/Waf1] in MDSCs is related to T cell suppression. Furthermore, although MDSCs reportedly promote angiogenesis[34] and the epithelial–mesenchymal transition (EMT)[35] to assist tumour growth, RNA-sequencing analysis showed that no angiogenesis- or EMT-inducing factors were downregulated in Mo-MDSCs from p16/p21-DKO mice compared with those from WT mice (Supplementary Data 1 and 2).

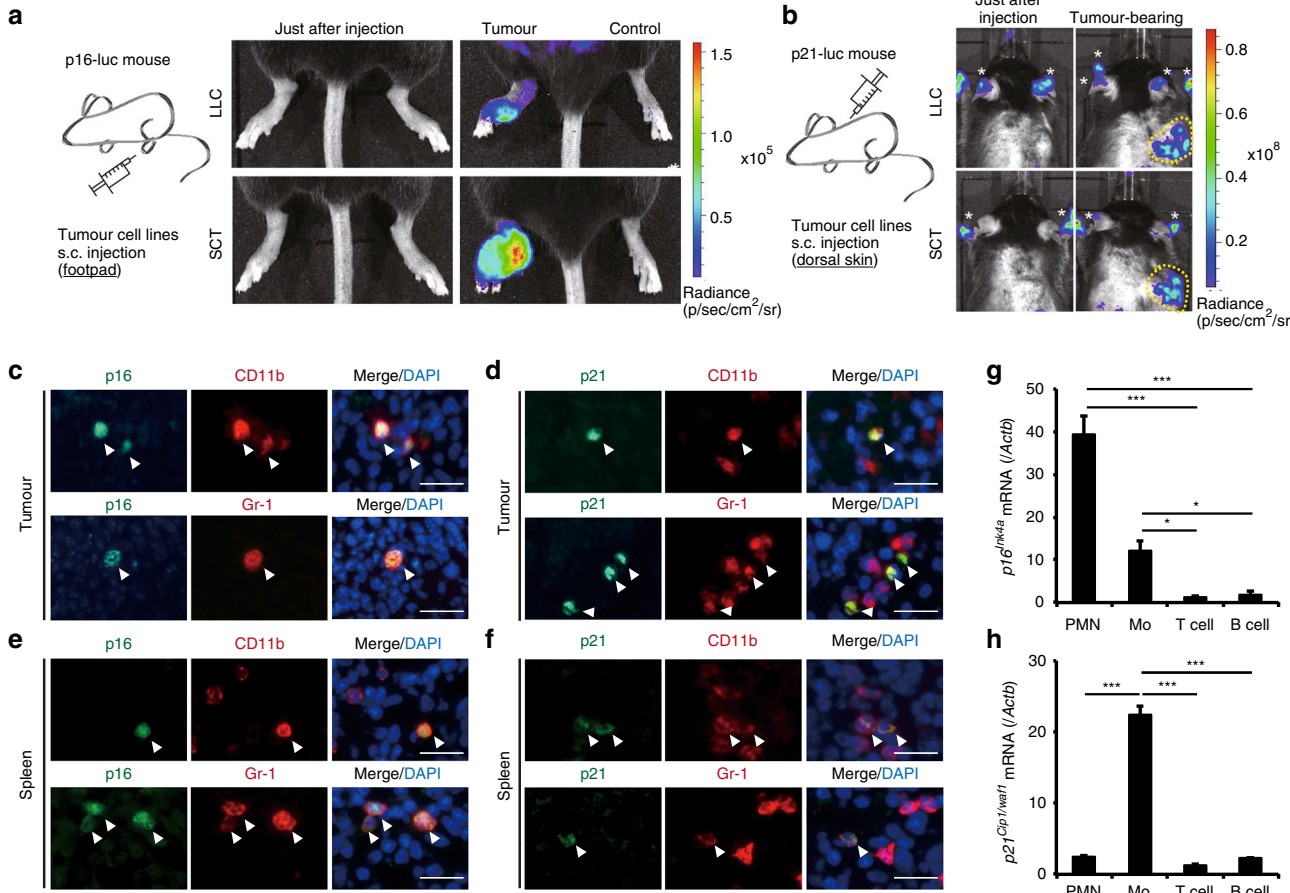

**Fig. 1** Tumour-induced MDSCs express p16[Ink4a] and p21[Cip1/Waf1]. **a** p16-luc mice were injected subcutaneously (s.c.) in the left footpad with LLC (3 weeks after injection; upper panels) or SCT (6 weeks after injection; lower panels) cells. The right footpad was injected with matrigel vehicle alone as a negative control. Representative bioluminescence images of mice just after tumour injection or tumour-bearing mice are shown. **b** Representative bioluminescence images of p21-luc mice that received s.c. injection of LLC (2 weeks after injection; upper panels) or SCT (4 weeks after injection; lower panels) cells in the dorsal area. Tumour areas are indicated by dotted circles; asterisks indicate signals from paws and ears. **c–f** Representative immunohistological images of tumour (**c**, **d**) and spleen (**e**, **f**) in an SCT-bearing mouse labelled with p16 (**c**, **e**) or p21 (**d**, **f**) antibodies (green) and either CD11b or Gr-1 antibodies (red). Arrowheads indicate p16- or p21-positive cells. Tissues were extirpated 3 weeks after SCT injection. Scare bars indicate 40 μm. **g**, **h** Expression of p16 (**g**) and p21 (**h**) mRNA in splenic PMN-MDSCs (PMN), Mo-MDSCs (Mo), T cells and B cells from SCT-bearing mice at 3 weeks after inoculation, as determined by qRT-PCR (n = 3); data are represented as the mean ± SEM. The statistical significance was determined by Student's t-test; *p < 0.05, ***p < 0.001

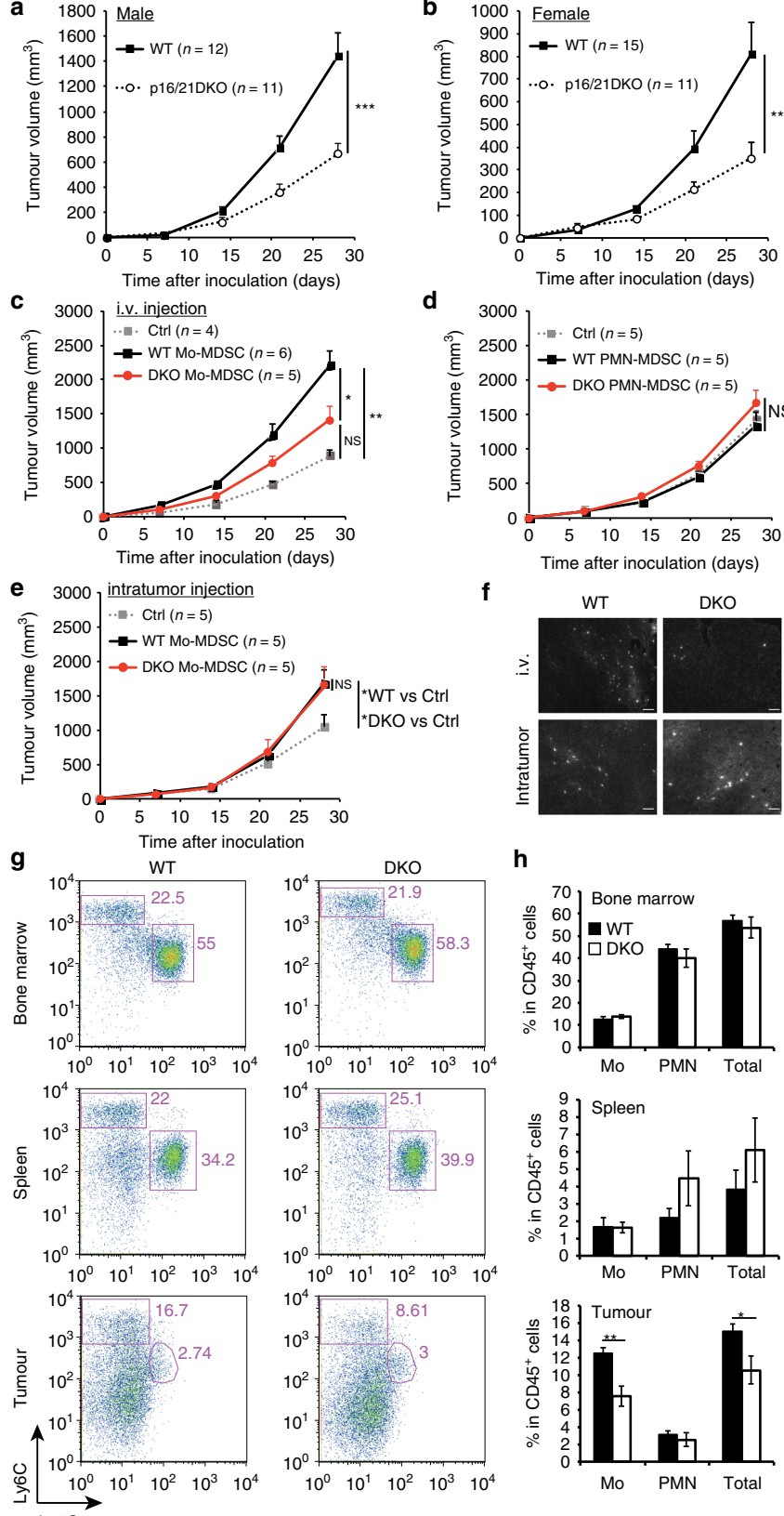

We thus next investigated whether the antitumoural phenotype in p16/p21-DKO mice is attributable to MDSC-mediated effects. Indeed, adoptive transfer experiments indicated that Mo-MDSCs purified from p16/p21-DKO mice were less effective in promoting tumour growth than those from WT mice (Fig. 2c). Consistent with a previous report[36], transferred PMN-MDSCs did not promote tumour growth regardless of the presence or absence of p16$^{Ink4a}$ and p21$^{Cip1/Waf1}$ expression (Fig. 2d). Collectively, these results strongly indicate that Mo-MDSCs exhibit differences in terms of tumour growth between WT and p16/p21-DKO mice. It should also be noted that p16$^{Ink4a}$- and p21$^{Cip1/Waf1}$-dependent facilitation of tumour growth was observed when Mo-MDSCs were intravenously transferred but not intratumorally transferred (Fig. 2c, e). These results suggest that the failure of Mo-MDSC infiltration into tumour allografts is likely to underlie defects in tumour-promoting activity in p16/p21-DKO mice. Certainly, more intravenously injected WT Mo-MDSCs were observed in the tumour than p16/p21-DKO Mo-MDSCs, but there was no difference in Mo-MDSC infiltration upon intratumoural injection (Fig. 2f). We additionally compared MDSC populations in tumour-bearing WT and p16/p21-DKO mice by flow cytometry. There were no differences between PMN- and Mo-MDSC populations in either the BM or spleen; however, fewer intratumoural Mo-MDSCs were obtained from p16/p21-DKO mice than from WT mice (Fig. 2g, h). Furthermore, more activated (CD44$^{high}$CD62L$^{low}$) CD8$^+$ T cells (Supplementary Fig. 4a, b), but fewer regulatory T cells, which are peripherally expanded by MDSCs[37,38], were present in the tumour-draining lymph nodes of p16/p21-DKO mice (Supplementary Fig. 4c, d). In addition, expression of another activation marker, CD69, was increased, and the population of cells positive for PD-1[39], an exhaustion marker, was decreased in intratumour CD8 T cells of p16/p21-DKO mice (Supplementary Fig. 4e–h). Together, these data demonstrate that Mo-MDSC localisation is important for local immunosuppression and consequent tumour development[17,40].

**CX3CR1 expression is reduced in Mo-MDSCs lacking p16 and p21.** We next sought to identify the mechanism underlying the inability of Mo-MDSCs to infiltrate tumour allografts in p16/p21-DKO mice. Chemotaxis is important for immune cell localisation; MDSC migration into tumours, which is dependent on chemokine receptors, affects tumour progression. We therefore attempted to identify chemokine receptors that were highly expressed in Mo-MDSCs but not in PMN-MDSCs, and that were additionally downregulated in p16/p21-DKO mice relative to WT mice. *Ccr2*, *Ccr5* and *Cx3cr1* were identified by RNA-sequencing analysis, and the differences in their expression levels were validated by qRT-PCR (Fig. 3a, b; Supplementary Fig. 5a–c). Flow cytometry analysis confirmed that CX3CR1 and CCR2, but not CCR5, were downregulated in splenic Mo-MDSCs lacking both p16$^{Ink4a}$ and p21$^{Cip1/Waf1}$ (Fig. 3c, d; Supplementary Fig. 5d, e).

CX3CR1 was also downregulated in the BM Mo-MDSCs lacking both p16$^{Ink4a}$ and p21$^{Cip1/Waf1}$, where progenitor cells differentiate into MDSCs and MDSCs undergo expansion (Fig. 3c, d). However, a significant difference in CX3CR1 expression in intratumour Mo-MDSCs between WT and p16/p21-DKO was not observed (Fig. 3c, d), although the number of intratumoural Mo-MDSCs in p16/p21-DKO mice was much smaller compared to WT mice, suggesting that CX3CR1 highly expressing Mo-MDSCs are concentrated in tumour site, regardless of the p16$^{Ink4a}$ and p21$^{Cip1/Waf1}$ gene status. This finding suggests that the inefficiency of Mo-MDSCs to infiltrate tumour allografts in p16/p21-DKO mice is likely attributable to the reduction of CX3CR1 expression in these mice.

**CX3CL1$^+$ tumours exhibit p16 and p21-dependent tumour growth.** To confirm the role of reduced CX3CR1 expression in the failure of Mo-MDSC infiltration into tumour allografts, we next assessed whether our findings were reproducible in other syngeneic tumour cell lines. Unexpectedly, LLC cells showed no differences in tumour growth between WT and p16/p21-DKO mice (Supplementary Fig. 6a). Notably, however, qRT-PCR analysis revealed that the expression level of CX3CL1 (also known as fractalkine), the sole CX3CR1 ligand, was much smaller in LLC cells than in SCTs (Fig. 4a; Supplementary Fig. 6b). On the other hand, the CCR2 ligands CCL2 and CCL7 were more highly expressed in LLC cells than in SCTs (Supplementary Fig. 6c, d). These results suggest that the tumour-suppressive effect of p16$^{Ink4a}$ and p21$^{Cip1/Waf1}$ is likely dependent on the levels of CX3CL1 expression in tumour cells.

Furthermore, slower growth of LLC tumour allografts in p16/p21-DKO mice was observed when CX3CL1 was ectopically expressed in LLC cells (Fig. 4a, b). Conversely, shRNA-mediated depletion of *Cx3cl1* in SCT cells significantly reduced tumour growth in WT syngeneic immunocompetent mice (Fig. 4a, c), but not in immunodeficient mice (Fig. 4d). As expected, the population of intratumoural Mo-MDSCs was reduced, whereas that of intratumoural CD8$^+$ T cells was increased by *Cx3cl1* knockdown relative to control SCTs (Fig. 4e, f). Consistently, CX3CL1 neutralisation antibody suppressed tumour progression (Fig. 4g), indicating that the CX3CL1–CX3CR1 pathway represents a potential therapeutic target. These data indicate that tumour growth acceleration induced by p16$^{Ink4a}$ and p21$^{Cip1/Waf1}$ is dependent on the CX3CL1–CX3CR1 pathway in tumours. It was previously reported that CX3CR1–CX3CR1 pathway regulates myeloid cell survival by inhibition of pro-apoptotic signals[41]; therefore, we analysed the apoptosis of Mo-MDSCs. However, there was no difference in the expression levels of the apoptotic inhibitory factor *Bcl2* between WT and DKO Mo-MDSCs (Supplementary Fig. 7a). The ratio of apoptotic Mo-MDSCs in spleen or tumour was similar between WT and DKO mice (Supplementary Fig. 7b). Thus attraction rather than

---

**Fig. 2** p16$^{Ink4a}$ and p21$^{Cip1/Waf1}$ in Mo-MDSCs promote tumour progression and the number of intratumoral Mo-MDSCs is decreased in p16/p21-DKO mice. **a, b** Tumour size in SCT-injected male WT and p16/p21-DKO mice (**a**) and female WT and p16/p21-DKO mice (**b**); the number of mice of each genotype is indicated in the data series. **c, d** Tumour size in WT mice subcutaneously injected with SCT cells simultaneously with adoptive transfer of WT or p16/p21-DKO Mo-MDSCs (**c**) and PMN-MDSCs (**d**) via the tail vein. **e** Tumour size in WT mice subcutaneously injected with a mixture of SCT cells and WT or p16/p21-DKO Mo-MDSCs. **f** Representative images of tumour slices of mice in which CSFE-labelled Mo-MDSCs from WT or DKO mice were injected intravenously (i.v.) or subcutanously (intratumour). Tumours were extirpated 21 days after injection. Signals indicate CSFE-labelled cells. Scale bars indicate 50 μm. **g** Representative flow cytometry plots of CD45$^+$CD11b$^+$-gated cells in BM (upper panels), spleen (middle panels), and tumour (lower panels) from WT and p16/p21-DKO mice 3 weeks after SCT cell injection. Cells were labelled with CD45, CD11b, Ly6C and Ly6G antibodies. **h** Mean populations of CD11b$^+$Ly6C$^{high}$Ly6G$^-$ (Mo) cells and CD11b$^+$Ly6G$^+$ (PMN) cells in the CD45$^+$-gated cells are shown in **f** (n = 5); data are presented as mean ± SEM. The statistical significance was determined by Student's *t*-test; *$p < 0.05$, **$p < 0.01$ and ***$p < 0.001$; NS not significant

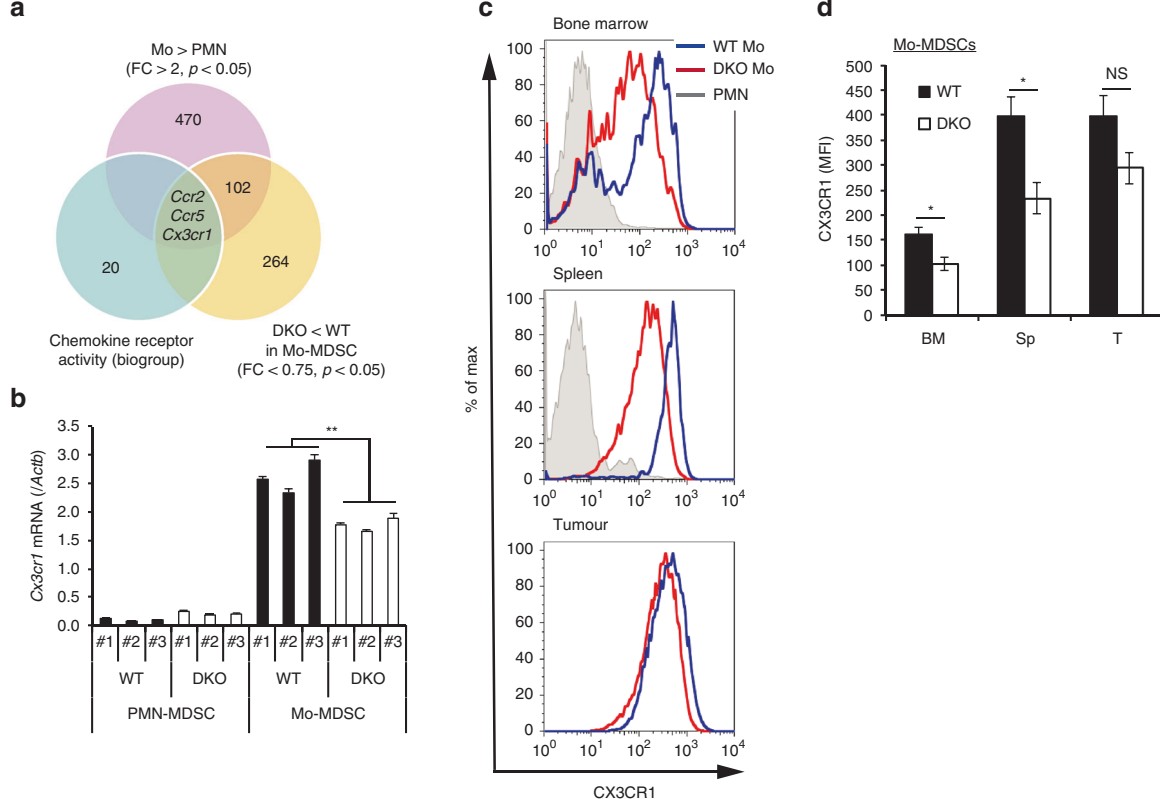

**Fig. 3** CX3CR1 expression is decreased in Mo-MDSCs in p16/p21-DKO mice. **a** Identification of the factor regulating p16/p21-dependent Mo-MDSC accumulation using RNA-sequencing data analysis; the Venn diagram shows the numbers of genes overexpressed in Mo-MDSCs compared with those in PMN-MDSCs (fold change >2, $p < 0.05$) and downregulated in DKO Mo-MDSCs compared with WT Mo-MDSCs (fold change <0.75, $p < 0.05$) categorised under chemokine receptor activity (biogroup). **b** Cx3cr1 mRNA expression in PMN- and Mo-MDSCs from WT or p16/p21-DKO mice ($n = 3$), as determined by qRT-PCR. **c** Representative flow cytometry histograms showing CX3CR1 expression in PMN- and Mo-MDSCs in BM (left), spleen (middle), and tumour (right) from WT and p16/p21-DKO mice 3 weeks after SCT cell injection. **d** Mean fluorescence intensity (MFI) of CX3CR1 expression in Mo-MDSCs shown in **c** ($n = 5$); data are presented as the mean ± SEM. The statistical significance was determined by Student's t-test; *$p < 0.05$ and **$p < 0.01$; NS not significant

survival of CX3CL1-dependent Mo-MDSCs is essential for immunosuppression and consequent SCT progression in vivo.

**IFN-γ enhances Mo-MDSC differentiation from BM cells.** We next examined the mechanisms underlying the effect of p16^Ink4a and p21^Cip1/Waf1 on Cx3cr1 expression in Mo-MDSCs. Purified Mo-MDSCs are unstable and rapidly differentiate into various myeloid cell types[16]; therefore, a stable Mo-MDSC culture was required for this step. MDSCs reportedly differentiate from BM cells in vitro by culturing in medium containing IL-6 and GM-CSF[42]; however, this method mainly induces differentiation into PMN-MDSCs but not Mo-MDSCs. It has been shown that the IFN-γ–STAT1 axis has an important role in the immunosuppressive ability of Mo-MDSCs[15,43]. Therefore, we tested whether IFN-γ is involved in the differentiation of MDSCs, especially into Mo-MDSCs. Flow cytometry analysis revealed a greater number of Mo-MDSCs in cultures containing IFN-γ than in those containing GM-CSF and IL-6 (Supplementary Fig. 8a). There were no differences in T cell suppression, the primary function of MDCS, between BM-derived cells cultured in the presence of GM-CSF, IL-6 and IFN-γ, and BM-derived MDSCs induced by GM-CSF with IL-6 (Supplementary Fig. 8b). In addition to Arg1 and Nos2, which are important for the immunosuppressive function of MDSCs, the MDSC marker genes S100a8 and S100a9[44,45] were strongly induced when cells were cultured in GM-CSF, IL-6 and IFN-γ (Supplementary Fig. 8c). On the basis of these results, we concluded that BM cells stimulated with GM-CSF, IL-6 and IFN-γ may be used as BM-derived Mo-MDSCs (BM-Mo-MDSCs).

**p16 and p21 regulate Cx3cr1 expression via SMAD3 signalling.** To obtain insights into the mechanisms by which p16^Ink4a and p21^Cip1/Waf1 regulate Cx3cr1 expression in Mo-MDSCs, BM-Mo-MDSCs were treated with the pan-CDK inhibitor flavopiridol[46,47] or the pan-E2F inhibitor HLM006474 (HLM)[48] (Fig. 5a). Interestingly, flavopiridol, but not HLM, increased Cx3cr1 expression (Fig. 5b), suggesting that p16^Ink4a and p21^Cip1/Waf1 regulate Cx3cr1 expression through the CDK, but not the E2F, pathway. Furthermore, although CDK4/6-specific inhibitors, LY2835219[49] and PD 0332991[50] have little effect on Cx3cr1 expression, CDK2-specific inhibitors, NU6027[51] and K03861[52], induce Cx3cr1 expression in BM-Mo-MDSCs (Fig. 5c). Consistent with cell cycle distribution of Mo-MDSCs (Supplementary Fig. 1a, upper), CDK kinase assay revealed that CDK2 activity of Mo-MDSCs in BM is higher than those in splenic or intratumour Mo-MDSCs, but a certain level of CDK2 activity was still existing in splenic Mo-MDSCs in WT mice and its activity was increased when p16^Ink4a and p21^Cip1/Waf1 genes were deleted (Fig. 5d).

We next attempted to identify the transcription factors that regulate Cx3cr1 expression under the control of CDK activity in Mo-MDSCs. Substantial reduction of tumour growth was observed in p16/p21-DKO mice but not in mice lacking either p16^Ink4a or p21^Cip1/Waf1 (single KO mice) (Fig. 2a, b; Supplementary Fig. 2), suggesting that both CDK2 and CDK4 associate

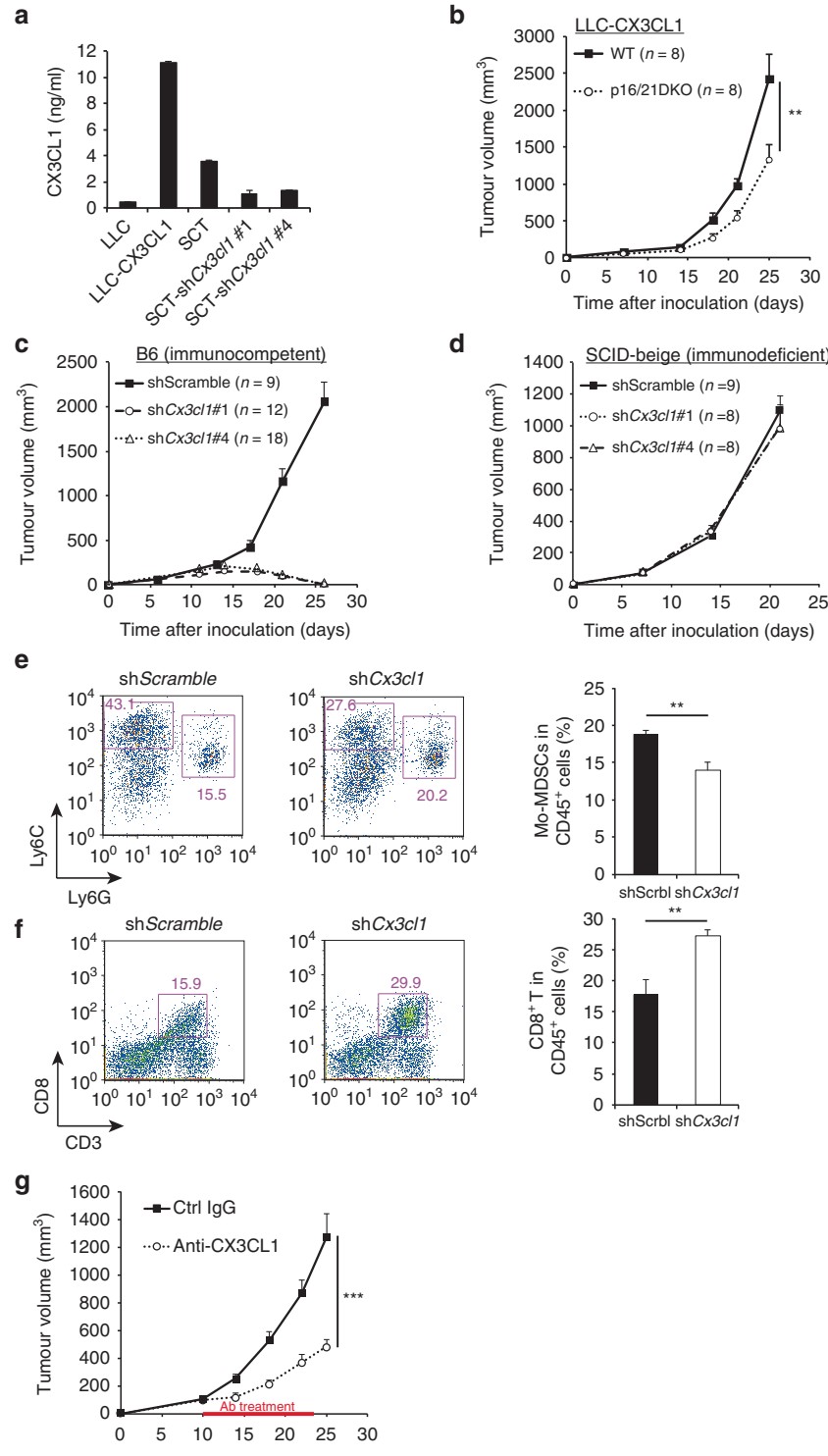

**Fig. 4** CX3CL1 expression in tumour cells is associated with p16/p21-dependent tumour progression. **a** Amount of soluble CX3CL1 in the indicated conditioned medium, as measured by enzyme-linked immunosorbent assay. Data were normalised to cell number. SCT cells were retrovirally transduced to express shRNA targeting *Cx3cl1* or a scrambled control shRNA. LLC-CX3CL1 indicates *Cx3cl1*-transduced LLC cells. **b** The growth curve of CX3CL1-overexpressing LLC cells in WT and p16/p21-DKO mice. **c, d** Tumour size in WT (**c**) and SCID-beige (**d**) mice injected with SCT-shScramble, SCT-sh*Cx3cl1*#1 and SCT-sh*Cx3cl1*#4. **e, f** Representative flow cytometry plots of CD45$^+$CD11b$^+$-gated (**e**) and CD45$^+$-gated (**f**) cells in tumours from WT mice 2 weeks after inoculation with *Cx3cl1*-knockdown or control SCT cells. Cells were labelled with CD45, CD11b, Ly6C and Ly6G (**e**) and CD45 and CD8 (**f**) antibodies. Graphs indicate CD11b$^+$Ly6C$^{high}$Ly6G$^-$ (Mo-MDSC) (**e**) and CD3$^+$CD8$^+$ (CD8$^+$ T cell) populations (**f**) among CD45$^+$ cells. **g** SCT growth curve in WT mice intraperitoneally injected every other day with anti-CX3CL1 antibody (4 μg per mouse) or isotype control rat IgG (4 μg per mouse) from day 10 to 23; data are presented as the mean ± SEM. The statistical significance was determined by Student's *t*-test; **$p < 0.01$ and ***$p < 0.001$; NS not significant

with this transcription factor in a complementary fashion. On the basis of this assumption, we identified SMAD3 as a candidate transcription factor using the STRING database (http://string-db.org). It has been reported that phosphorylation of the SMAD3 linker region by CDK2 and CDK4 inhibits SMAD3 transcriptional activity[53–55]. Indeed, the linker phosphorylation (Ser-213) in SMAD3 was increased in BM and splenic Mo-MDSCs from

p16/p21-DKO mice compared with those from WT mice (Fig. 5e). Notably, the levels of CX3CR1 expression were inversely correlated with those of phosphorylated SMAD3 (Figs. 3c and 5e). These data suggest that p16[Ink4a] and p21[Cip1/Waf1] have important roles in controlling CX3CR1 expression for the property of chemotaxis in Mo-MDSCs, by tuning CDK2 activity during their maturation.

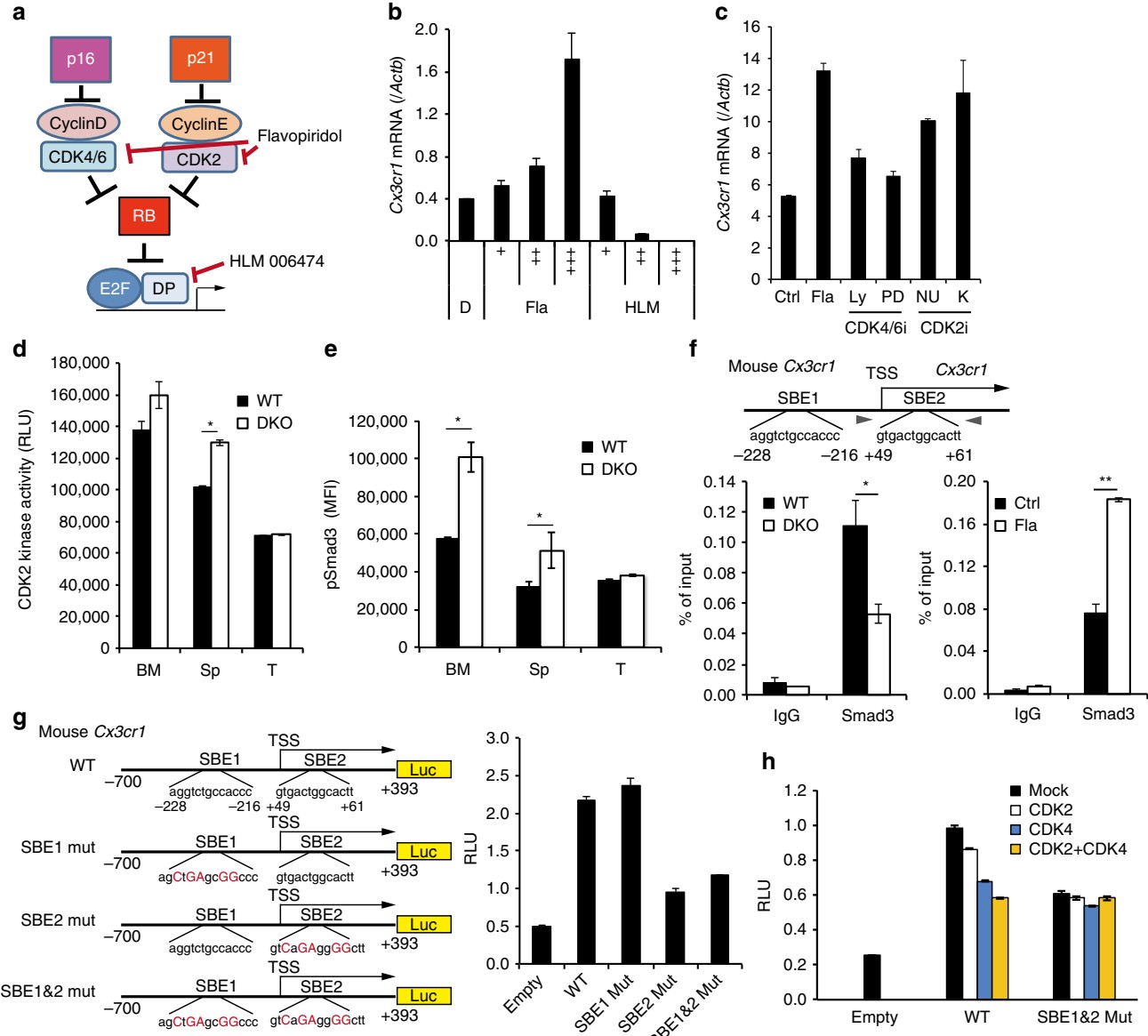

**Fig. 5** CDK-mediated SMAD3 phosphorylation inhibits *Cx3cr1* expression. **a** Diagram of the CDKI-CDK-RB-E2F pathway and sites of action of chemical inhibitors. **b** Expression of *Cx3cr1* mRNA in BM-Mo-MDSCs treated with indicated doses of flavopiridol (Fla), HLM006474 (HLM) or dimethyl sulfoxide (DMSO) control (D). **b** "+" indicates drug concentration: Fla; + (4 nM), ++ (20 nM), +++ (100 nM). HLM; + (2 nM), ++ (10 nM), +++ (50 nM). **c** Expression of *Cx3cr1* mRNA in BM-Mo-MDSCs treated with flavopiridol (Fla; 50 nM), LY2835219 (Ly; 5 nM), PD 0332991 (PD; 5 nM), NU6027 (NU; 5 μM), K03861 (K; 250 nM) or DMSO (Ctrl). **d** Kinase activity of CDK2 against recombinant SMAD3. CDK2 was immunoprecipitated from lysates of Mo-MDSCs purified from bone marrow (BM), spleen (Sp) and tumour (T). **e** Flow cytometric analysis of phosphorylated Smad3 (S213). Graphs indicate mean fluorescence intensity (MFI) of indicated protein expression in Mo-MDSCs (n = 4). **f** Smad3 binding to the *Cx3cr1* promoter region was evaluated by ChIP. Lysates were prepared from BM-Mo-MDSCs derived from WT or p16/p21-DKO mice (left). WT BM-Mo-MDSCs were treated with 100 nM flavopiridol (Fla) for 24 h (right). Two SBEs are located around the transcriptional start site (TSS) of the mouse *Cx3cr1* gene. Arrowheads indicate primer positions. **g**, **h** Schematic representation of mouse *Cx3cr1* promoter reporter constructs; two SBEs (SBE1 and SBE2) are shown in the sequence, and mutation sites are indicated by red uppercase letters. THP-1 cells were transfected with the indicated reporter plasmids and pCMV-Renilla (**g**). Where indicated, cells were also co-transfected with expression plasmids encoding CDK2, CDK4 or both (**h**). At 48 h after transfection, the luciferase activities were measured. Data are representative of three biological replicates and presented as the mean ± SEM. The statistical significance was determined by Student's t-test; *p < 0.05 and **p < 0.01

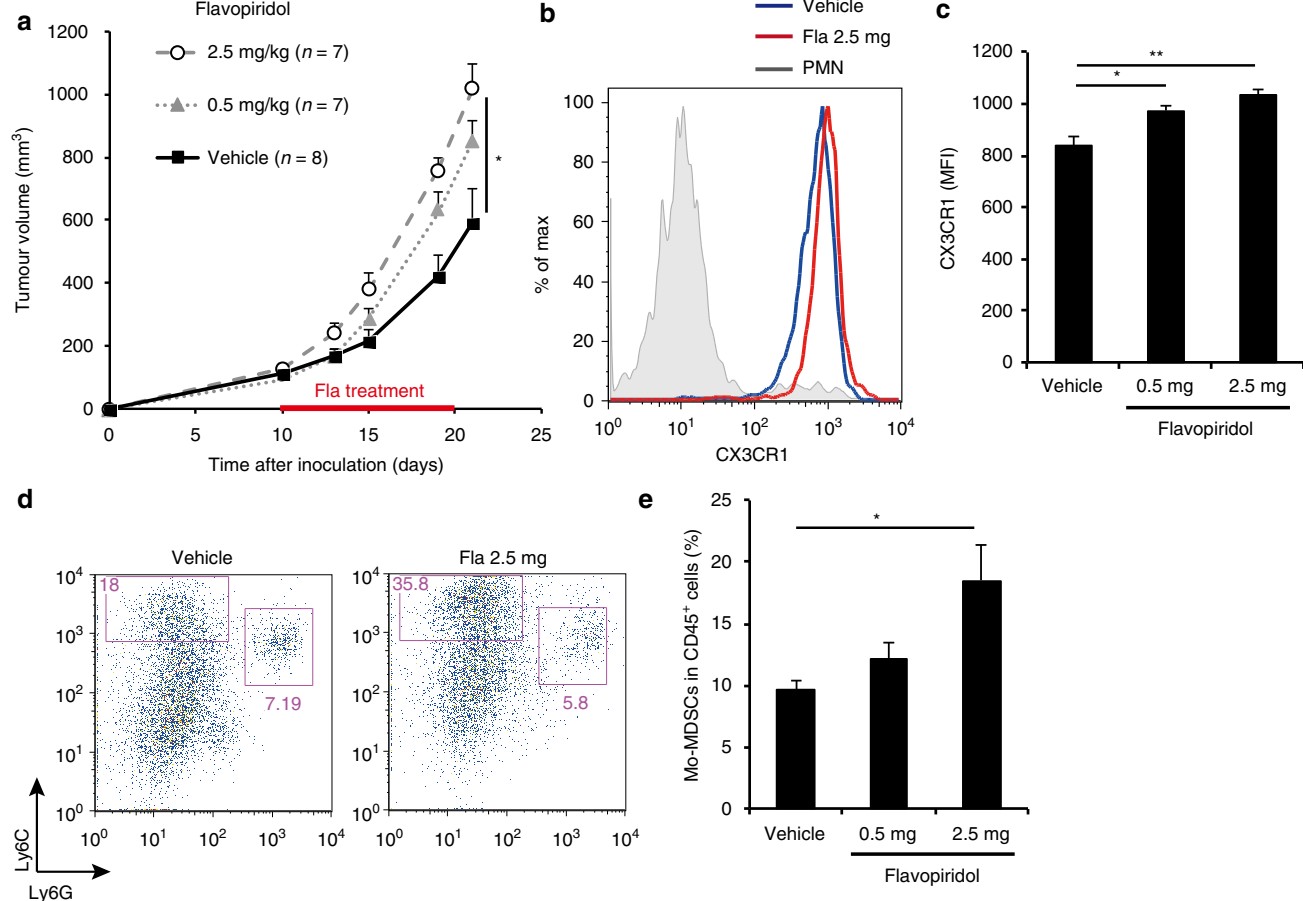

**Fig. 6** Flavopiridol accelerates tumour growth *in vivo*. **a** SCT growth curve in WT mice administered flavopiridol (2.5 or 0.5 mg/kg) or vehicle (control) once daily, from day 10 to day 20. **b** Representative flow cytometry histograms showing CX3CR1 expression in splenic Mo-MDSCs from WT mice 10 days after flavopiridol administration. **c** Mean fluorescence intensity (MFI) of CX3CR1 expression shown in **b** (*n* = 4). **d** Representative flow cytometry plots of CD45$^+$CD11b$^+$-gated cells in tumours from WT mice after flavopiridol administration; cells were labelled with CD45, CD11b, Ly6C and Ly6G antibodies. **e** Fraction of CD11b$^+$Ly6C$^{high}$Ly6G$^-$ cells among CD45$^+$-gated cells shown in **d** (*n* = 5). Data are presented as the mean ± SEM. The statistical significance was determined by Student's *t*-test; *$p < 0.05$ and **$p < 0.01$

We next confirmed whether the change of linker phosphorylation level in SMAD3 affects *Cx3cr1* expression directly. Two SMAD-binding elements, SBE1 and SBE2, were present in the *Cx3cr1* promoter region. A chromatin immunoprecipitation (ChIP) assay showed that SMAD3 binding to SBE2 was downregulated in p16/p21-DKO and upregulated by treatment with a CDK inhibitor (Fig. 5f). SMAD3 activation then stimulated *Cx3cr1* promoter activity via binding to SBE2 (Fig. 5g) and ectopic expression of CDK2 inhibited SBE-mediated *Cx3cr1* promoter activity (Fig. 5h). Together, these data strongly suggest that inactivation of CDK2 induce *Cx3cr1* expression via SMAD3 activation.

**Flavopiridol enhances SCT growth in vivo.** CDK inhibitors such as flavopiridol, which is used to treat leukaemia[46], have attracted recent attention as a result of their potential as anticancer agents. However, the present results suggest that CDK inhibition may accelerate tumour growth by enhancing Mo-MDSC infiltration into tumours expressing CX3CL1 in vivo. To explore this possibility, we generated SCT cells expressing SV40 large T antigen (SCT-LT), in which the retinoblastoma tumour suppressor (Rb)-pathway is inactivated, to elicit resistance against CDK inhibitors (Supplementary Fig. 9a, b). The growth of SCT-LT was substantially increased in mice treated with flavopiridol,

accompanied by enhanced CX3CR1 expression in Mo-MDSCs and increased accumulation of Mo-MDSCs in tumours (Fig. 6a–e). These data, in conjunction with the notion that a certain proportion of human cancers contains disruption of the Rb-pathway, strongly suggest that treatment with CDK inhibition may promote tumour development, depending on the level of expression of CX3CL1 and the Rb status in the tumour.

## Discussion

To our knowledge, the present work is the first to report that p16$^{Ink4a}$ and p21$^{Cip1/Waf1}$ CDK inhibitors, which are considered cell-intrinsic tumour suppressors, in fact promote tumour development, as demonstrated in the present allograft mouse model. Both p16$^{Ink4a}$ and p21$^{Cip1/Waf1}$ are expressed in Mo-MDSCs and facilitate infiltration of Mo-MDSC into the tumour microenvironment by upregulating CX3CR1 expression and conferring resistance to anti-tumour immune mechanisms (Figs. 1–5; Supplementary Fig. 10a). Notable, moreover, administration of flavopiridol was found to accelerate the development of orthotopically transplanted SCT-LT cells in syngeneic mice (Fig. 6; Supplementary Fig. 10b). These results, together with previous reports showing that high p16$^{INK4A}$ expression in the tumour stroma was strongly associated with high risk of recurrence[56] or poor survival[57] in some types of cancer, suggest that

there might be the negative side effects of CDK inhibitor drugs used for cancer treatment.

Regarding the role of p16[Ink4a] in myeloid cells, it is known to promote M1 polarisation, which reflects an inflammatory state for macrophages[58]. Indeed, the present study also showed *Arg1* upregulation in p16/p21-DKO Mo-MDSCs (Supplementary Fig. 3d). It is generally believed that M2 skewing promotes tumour growth. Therefore, it is most likely that the decreased tumour growth in p16/p21-DKO mice is caused by MDSC alteration rather than defective macrophage polarisation.

Although the role of CDK2 and CDK4 (CDK2/4) in cell cycle control is well established, there is increasing evidence that CDK2/4 also have cell-cycle independent roles in various non-proliferating cells[55]. In the present study, we revealed a novel function of CDK2 in regulating CX3CR1 expression via phosphorylation and inactivation of SMAD3 in Mo-MDSCs (Fig. 5). On the basis of our comparative data of MDSCs in bone marrow, in spleen and in tumour, we propose a possible scenario whereby p16/p21 regulates Mo-MDSCs function during their maturation. Once Mo-MDSCs leave BM, they start to mature, terminate its proliferation and exert their own character, for example chemotaxis and immunosuppression. During this maturation process, CDK2 activity is sustained, yet not enough to drive cell cycle, and tuned by p16[Ink4a] and p21[Cip1/Waf1] to lead the expression of CX3CR1 for chemotaxis (Supplementary Fig. 1a; Figs. 3c, d and 5d, e).

The chemotaxis of immune cells contributes to important aspects of the tumour microenvironment[59]; in particular, the use of immune checkpoint inhibitors as therapeutic agents, which has achieved significant results in cancer therapy, has revealed the importance of tumour-infiltrating immune cells for predicting the response to checkpoint blockade[60,61]. Note that the absence of CX3CL1 expression in pre-treatment tumours[60] is reportedly correlated with the response to the anti-PD-L1 antibody. We speculate that high CX3CL1 expression might recruit Mo-MDSCs to tumour, thereby increasing PD-L1-independent immunosuppression. However, the role of the CX3CL1–CX3CR1 axis in anti-tumour immunity is highly controversial. For instance, the correlation between CX3CL1 expression and the poor prognosis of cancer patients varies depending on the type of cancer: in colorectal cancer and gastric adenocarcinomas, high CX3CL1 expression is associated with better prognosis[62,63]; however, the opposite outcome is observed in breast cancer and hepatocellular carcinoma[64,65]. Using SCT cells, which are derived from the skin of *p16[Ink4a]* knockout mice and are highly expressing CX3CL1, we showed that treatment with anti-CX3CL1 exerted significant suppressive effects against tumour progression, which has not been previously demonstrated in vivo (Fig. 4g). We therefore suggest that CX3CL1 represents a novel molecular target for cancer therapy, especially in combination with CDK inhibitors (Supplementary Fig. 10b).

In tumour microenvironment, cancer-associated fibroblasts (CAFs) and pre-cancerous cells are shown to exhibit senescence-like phenotypes including SASPs. For example, it was recently reported that some CAFs and pre-cancerous hepatocytes recruit MDSCs via IL-6 and CCL2-mediated pathway, respectively, thereby protecting cancer cells from immune cells[66,67]. These results, together with our findings that p16[Ink4a] and p21[Cip1/Waf1] enhance Mo-MDSC chemotaxis, led us to speculate that the formation of the tumour microenvironment may have a synergistic effect on protumoural MDSC recruitment

The role of p16[Ink4a] in the development of cellular senescence is well known; furthermore, p16[Ink4a] has been the most widely accepted marker for senescent cells for over two decades[1,11,12,68]. Notably, moreover, a recent study reported that elimination of p16[Ink4a]-expressing senescent cells from mice prolonged lifespan

and alleviated ageing related diseases including cancer[69], indicating that the elimination of senescent cells would be beneficial for homoeostasis, especially in aged animals. However, because MDSCs express high levels of p16[Ink4a] without any noticeable senescent phenotype, the results of eliminating p16[Ink4a] expressing cells must be interpreted with caution, as previously suggested by Baker et al.

In summary, our results reveal a novel function of p16[Ink4a] and p21[Cip1/Waf1] CDK inhibitors in MDSCs, thus expanding our understanding of the mechanisms by which the chemotaxis of MDSCs is regulated. A greater understanding of the molecular mechanisms linking these CDK inhibitors to the chemotaxis of immune cells should therefore provide valuable novel strategies to control tumorigenesis.

## Methods

**Mice**. WT C57BL/6J mice were purchased from Charles River Laboratories Japan and CLEA Japan. SCID-Beige (CB17.Cg-Prkdc[scid]Lyst[bg-J]/CrlCrlj) mice were purchased from Charles River Laboratories Japan. p16/p21-DKO mice were generated as previously reported[9]. p16-luc mice[11] and p21-luc mice[10] were generated as previously reported and backcrossed with C57BL/6 mice for at least 8 generations. All animals were maintained according to protocols approved by the Committee for the Use and Care of Experimental Animals of the Japanese Foundation for Cancer Research (Tokyo, Japan) and the Research Institute for Microbial Diseases, Osaka University (Osaka, Japan).

**Cell culture and tumour inoculation**. The SCT line was prepared from a DMBA- and TPA-treated p16 KO mouse[9]. LLC (RCB0558) cell was provided by RIKEN BioResource Center through the National BioResource Project of MEXT, Japan. SCT and LLC cells were cultured in Dulbecco's modified Eagle medium (08458–45; Nacalai Tesque) supplemented with 10% foetal bovine serum (FBS; Gibco/Thermo Fisher Scientific), 100 U/ml penicillin, and 100 µg/ml streptomycin (Sigma-Aldrich). All cell lines were cultured at 37 °C in 5% $CO_2$. For injection, cultured cells were collected in logarithmic growth phase (40–70% confluence). Mice (12- to 20-weeks-old) were anaesthetised with pentobarbital, and their flanks were subcutaneously injected with the following cells: SCT ($1 \times 10^6$), LLC ($1 \times 10^5$), SCT-LT ($1 \times 10^6$), LLC-CX3CL1 ($1 \times 10^5$) or SCT-shCx3cl1 ($1 \times 10^6$). All cell lines were resuspended in 50% Matrigel (Corning). It was confirmed that cell lines had no mycoplasma contamination before inoculation into mice. Measurements of tumour volume were a performed in blinded set-up.

**Bioluminescence imaging**. Isoflurane-anaesthetised mice were intraperitoneally injected with 75 mg/kg D-luciferin substrate (Wako Pure Chemical Industries) (30 mg/ml in saline) and imaged using an IVIS Lumina system (Caliper Life Sciences). Luminescence was determined to begin 5 min after substrate administration. Female p16-Luc mice were subcutaneously injected in the footpad with LLC ($2.5 \times 10^5$) or SCT ($1 \times 10^6$) cells. After 3 weeks (LLC) or 10 weeks (SCT), tumour-bearing mice were subjected to bioluminescence imaging. p21-Luc mice were subcutaneously injected on the dorsal side with LLC ($4 \times 10^5$ cells) or SCT ($1 \times 10^6$ cells). After 2 weeks (LLC) or 3 weeks (SCT), the mice were subjected to bioluminescence imaging.

**Immunohistochemistry**. Tissue specimens were embedded in optimal cutting temperature compound (Sakura Finetek) and frozen in liquid nitrogen. Cryostat sections of 5-µm thickness were fixed with 4% paraformaldehyde (Nacalai Tesque) for 30 min, blocked in 1% bovine serum albumin (Sigma-Aldrich) and 10% FBS, and stained with the antibodies listed in Supplementary Data 4. Non-specific labelling of Fc receptors was prevented by incubating the cells with anti-CD16/CD32 antibody (1:500, 93; BioLegend).

**MDSC isolation**. PMN-MDSCs (Ly6G+) and Mo-MDSCs (Ly6G−Gr-1+) were isolated from the BMs, spleens, or tumours of SCT-bearing mice 3 weeks after injection, using the Myeloid-Derived Suppressor Cell Isolation kit for mice (Miltenyi Biotec) according to the manufacturer's instructions, with all steps performed at 4 °C. The purity of MDSC subfractions was typically higher than 95%.

**Quantitative real-time RT-PCR**. For analysis of gene expression, total RNA was extracted using the mirVana RNA Isolation kit (Ambion/Thermo Fisher Scientific) and cDNA was synthesised using the PrimeScript RT Reagent kit (Takara Bio). Quantitative real-time RT-PCR (qRT-PCR) was performed using SYBR Premix Ex Taq (Takara Bio) on a Prism 7900HT system (Applied Biosystems). Primers used are listed in Supplementary Data 3.

**Flow cytometry analysis**. Cell suspensions were prepared from spleen or lymph node tissue by sieving and gentle pipetting. To prepare intratumoural cell suspensions, minced tumours were agitated in Hank's balanced salt solution containing 1 mg/ml collagenase IV, 100 μg/ml hyaluronidase and 2 U/ml DNASe IV (Sigma-Aldrich) at 20–25 °C for 2 h, and strained through a 70-μm mesh (Greiner Bio-One). Non-specific labelling of Fc receptors was prevented by incubating the cells with anti-CD16/CD32 antibody (1:500, 93; BioLegend). Intracellular staining of FoxP3 and intranuclear staining were performed with the FOXP3 Fix/Perm Buffer Set (BioLegend) and True-Nuclear Transcription Factor Buffer Set (BioLegend), respectively, according to the manufacturer's protocol. The antibodies used are listed in Supplementary Data 4. Flow cytometry was performed on a FACSAria instrument with FACSDiva software (BD Biosciences) or Attune NxT (Thermo Fisher Scientific), and data were analysed with FlowJo v9 or v10 software (Tree Star).

**Cell cycle analysis**. For flow cytometry-based cell cycle analysis, cells were labelled with antibodies against cell surface antigens, fixed with 4% paraformaldehyde (Nacalai Tesque) for 15 min, and incubated in 10 μg/ml 4′,6-diamidino-2-phenylindole (Dojindo)/0.1% Triton X-100 (Nacalai Tesque) in phosphate-buffered saline (PBS) for 30 min at 20–25 °C.

**Proliferation assay**. For the in vitro proliferation assay, purified splenic Mo-MDSCs, PMN-MDSCs and T cells were stained with 5 μM carboxyfluoroscein diacetate succinimidyl ester (CFSE; eBioscience/Affymetrix) according to the manufacturer's instructions. Overall, $1 \times 10^5$/ml CFSE-labelled Mo- and PMN-MDSCs are cultured in 10% FBS-containing Roswell Park Memorial Institute (RPMI)1640 medium (30264–85; Nacalai Tesque) supplemented with 20 ng/ml mouse GM-CSF (Peprotech) for 48 h. Dividing cells were identified as those showing diluted CFSE signals relative to unstimulated T cells. For the in vivo 5-ethynyl-2′-deoxyuridine (EdU) incorporation assay, purified Mo-MDSCs, PMN-MDSCs and T cells were stained with 5 μM CFSE. SCT-bearing mice were given drinking water containing 1 mg/ml EdU (Thermo Fisher Scientific) for 72 h just after CFSE-labelled cell injection ($1 \times 10^6$/mouse)[70]. EdU incorporation was detected using a Click-iT EdU Alexa Fluor 647 Flow Cytometry Assay Kit (Thermo Fisher Scientific). The proliferation rate was calculated as the number of EdU-positive cells among CFSE-positive cells.

**Senolytic drug treatment and apoptosis assay**. For in vitro experiments, young mouse embryonic fibroblasts (MEF) were cultured in 3% $O_2$. Senescent MEFs were prepared by treatment with 15 ng/ml doxorubicin hydrochloride (Sigma-Aldrich) in 20% $O_2$ for 7 days. Cells were treated with ABT-263 (LKT Laboratories, Inc.) for 72 h. GM-CSF, IL-6 and IFN-γ (Peprotech) were added to the culture medium during treatment of BM-Mo-MDSCs with ABT-263. Apoptotic cell staining was performed with the PE Annexin V Apoptosis Detection Kit with 7-AAD (BioLegend), according to the manufacturer's instructions. Apoptotic cells were detected by FACSCalibur (BD Biosciences). For in vivo experiments, wild-type mice were treated with control vehicle (10% ethanol, 30% polyethylene glycol 400 and 60% Phosal 50 PG; Phospholipid Gmbh), or 50 mg/kg/d ABT-263 from day 21 to day 27 after SCT inoculation. At day 28, peripheral blood was collected and stained for Mo-MDSCs (CD11b⁺Ly6C^{high}Ly6G⁻) and PMN-MDSCs (CD11b⁺Ly6C^{int}Ly6G⁺) among CD45⁺ cells.

**T-cell proliferation suppression assay**. The antigen-dependent T cell activation assay[29] was performed to assess antigen-dependent T-cell suppression. DC2.4 dendritic cells and RF33.70 OVA-SIINFEKL specific T-T hybridoma cells were provided by K. L. Rock (University of Massachusetts Medical School, Worcester, MA, USA)[27,28]. Briefly, $1 \times 10^3$ DC2.4 cells were cultured overnight in a 96-well plate and then incubated with OVA-SIINFEKL peptide (MBL International) for 24 h. OVA-pulsed cells were treated with 10 μg/ml mitomycin C (Sigma-Aldrich) for 30 min and washed twice with PBS. DC2.4 cells were co-cultured with $1 \times 10^4$ RF33.70 cells and purified splenic PMN- or Mo-MDSCs ($1 \times 10^4$, $5 \times 10^3$, $2.5 \times 10^3$ or $1.25 \times 10^3$ cells) for 20 h. The amount of IL-2 released into the culture medium was measured with a murine IL-2 Mini TMB ELISA Development Kit (Peprotech) according to the manufacturer's instructions. Antigen-independent T cell receptor stimulation was performed as follows: T cells were isolated from the spleens of WT C57/BL6J mice 3 weeks after injection using the Pan-T Cell Isolation Kit II for mouse (Miltenyi Biotec). Isolated T cells were stained with 5 μM CFSE. CFSE-labelled T cells were stimulated with Dynabeads Mouse T-Activator CD3/CD28 (Veritas Life Science) and co-cultured with purified splenic Mo-MDSCs in 96-well round-bottomed plates (BM Equipment). After 4 days, cells were labelled with antibodies against CD4 or CD8 (BioLegend), and the CFSE signal from live CD4⁺- or CD8⁺-gated cells was analysed by flow cytometry. Dividing cells were identified as those showing diluted CFSE signals relative to unstimulated T cells.

**Adoptive cell transfer**. PMN- and Mo-MDSCs were purified from the spleens of SCT-bearing WT and p16/p21-DKO mice 3 weeks after SCT injection. For intravenous transfer, 8- to 10-week-old WT C57BL/6 J mice were subcutaneously injected with $1 \times 10^6$ SCT cells in Matrigel (Corning), followed by injection of $2 \times 10^6$ PMN- or Mo-MDSCs into the tail vein. For subcutaneous transfer, 10-week-old WT C57BL/6J mice were subcutaneously injected with a mixture of $1 \times 10^6$ SCTs and $2 \times 10^6$ Mo-MDSCs in 50% Matrigel. For observation of injected Mo-MDSCs, $2 \times 10^6$ Mo-MDSCs were stained with 2 μM CFSE and then injected intravenously or subcutaneously.

**RNA-sequencing**. PMN- and Mo-MDSCs were purified from the spleens of SCT-bearing WT and p16/p21-DKO mice. RNA isolation was performed using the mirVana RNA Isolation kit. Library preparation was performed using a TruSeq stranded mRNA sample prep kit (Illumina) according to the manufacturer's instructions. Sequencing was performed on an Illumina HiSeq 2500 platform in a 75-base single-end mode. Illumina Casava1.8.2 software was used for base calling. Sequenced reads were mapped to the mouse reference genome sequences (mm10) using TopHat v2.0.13 in combination with Bowtie2 ver. 2.2.3 and SAMtools ver. 0.1.19. The fragments per kilobase of exon per million mapped fragments (FPKMs) were calculated using Cuffnorm version 2.2.1.

**Viral infection and establishment of stable cell lines**. LLC and SCT cells were infected with retrovirus encoding mouse *Cx3cl1* (in pMarX-puro) and mouse *Cx3cl1* shRNA (in pRetrosuper-puro) or SV40LT antigen (in pMarX-puro), respectively. Infected cells were selected in 5 μg/ml puromycin-containing medium for 7 days, and the selected cells were maintained in medium containing 1 μg/ml puromycin (Wako Pure Chemical Industries).

**In vivo drug administration**. For anti-CX3CL1 experiments, mice were administered either anti-CX3CL1 rat IgG (MAB571; R&D Systems) or isotype anti-rat IgG (MAB006; R&D Systems) (4 μg/mouse by intraperitoneal injection) once daily for 14 days starting on day 10 post-SCT inoculation. For CDK inhibition, mice were given daily intraperitoneal injections of flavopiridol HCl (Selleckchem) (2.5 or 0.5 mg/kg/day) or 1% dimethyl sulphoxide (DMSO) in PBS as a vehicle control for 11 days starting on day 10 post-SCT-LT inoculation.

**BM-Mo-MDSC preparation**. The protocol for BM-Mo-MDSC preparation was modified from that used in a previous report. Femurs were removed from C57BL/6 mice and the BM was flushed. Red blood cells were lysed with ammonium chloride. BM cells were adjusted to a concentration of $1 \times 10^6$/ml in 10% FBS-containing RPMI1640 medium supplemented with 20 ng/ml mouse GM-CSF, 40 ng/ml mouse IL-6 and 2 ng/ml mouse IFN-γ (Peprotech). Cells were maintained at 37 °C in a 5% $CO_2$- and 3% $O_2$-humidified atmosphere for 4–5 days. Non-adherent and loosely adherent cells were collected by gentle pipetting. In some experiments, BM-Mo-MDSCs were treated with flavopiridol HCl (Selleckchem) or HLM (Tocris Bioscience) along with GM-CSF, IL-6, and IFN-γ on day 4, and collected 24 h later.

**In vitro kinase assay**. One million Mo-MDSCs purified from bone marrow, spleen and tumour were lysed by cell lysis buffer (Cell Signaling Technology), and immunoprecipitated by anti-CDK2, and Protein A Dynabeads (Thermo Fisher Scientific). The antibodies are listed in Supplementary Data 4. For kinase reaction, precipitated CDK2 was incubated in 20 mM Tris-HCl (pH7.5) containing 5 ng/μl of recombinant SMAD3 (Sigma-Aldrich), 100 μM ATP (Promega), 0.1 mg/ml BSA, 1 mM DTT and 40 mM MgCl₂ at 30 °C for 2 h. ATP to ADP conversion during kinase reaction was measured by ADP-Glo Kinase Assay (Promega) and Powerscan HT (DS Pharma Biomedical).

**ChIP analysis**. ChIP analysis was performed using EZ-ChIP (Millipore), Dynabeads Protein G (Thermo Fisher Scientific) and anti-Smad3 antibody (Abcam) or normal rabbit IgG (Cell Signaling Technology) as a negative control. Information on the antibodies used may be found in Supplementary Data 4. Immunoprecipitated DNA was quantified by qRT-PCR. For information on primers used, see Supplementary Data 3.

**Luciferase reporter assay**. The mouse *Cx3cr1* gene promoter region (from −700 to +397) was amplified from the genome of a C57BL/6J mouse. PCR mutagenesis for 2 SBEs was performed using the primers listed in Supplementary Data 3. The amplified fragments were inserted into the pGL3basic vector (Promega). cDNAs for human CDK2 and CDK4 were obtained by PCR and subcloned into the pcDNA3. THP-1 (JCRB0112.1) cells were provided by RIKEN BioResource Center through the National BioResource Project of MEXT, Japan and cultured in RPMI1640 medium with 10% FBS, 10 mM HEPES buffer, 100 μM sodium pyruvate (Gibco/Thermo Fisher Scientific), 100 U/ml penicillin, 100 μg/ml streptomycin and 50 mM 2-mercaptoethanol (Wako Pure Chemical Industries). THP-1 cells were transfected with plasmids, using Amaxa Cell Line Nucleofector Kit V and Nucleofector I Device (Lonza); 48 h later, luciferase activity was measured with the Dual-Luciferase Reporter Assay System (Promega) according to the manufacturer's instructions. Reporter activity was normalised to that of Renilla luciferase derived from the co-transfected pCMV-Renilla vector. Data are representative of three independent experiments.

**Western blotting**. Cells were lysed in Laemmli's sample buffer. The samples were subjected to SDS-polyacrylamide gel electrophoresis followed by immunoblot analysis with the indicated antibodies and appropriate secondary antibodies conjugated with horseradish-peroxidase. Bound antibodies were visualised by chemiluminescence following incubation with Chemi-Lumi One L HRP Substrate (Nacalai Tesque) or Amersham ECL Prime (GE Healthcare). For information on the antibodies used, see Supplementary Data 4.

**Statistical analysis**. Student's *t*-test (two-tailed) was performed to evaluate the significance of differences between two groups.

**Data availability**. RNA-sequencing data have been deposited in the NCBI's Gene Expression Omnibus database (GEO GSE93359). All other remaining data are available within the article and supplementary files, or available from the authors upon request.

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

## Acknowledgements

We acknowledge the NGS core facility of the Genome Information Research Center at the Research Institute for Microbial Diseases of Osaka University for support with RNA-sequencing and data analysis. We thank K. Rock (University of Massachusetts Medical School) and M. Ito (The Jikei University School of Medicine) for providing RF33.70 and DC2.4 cells. We also thank the members of the Hara laboratory for helpful discussion during the preparation of this manuscript. This work was supported by Japan Society for the Promotion of Science (JSPS) KAKENHI Grant Numbers JP16K19064 and JP14J10280 and 26250028, and the Japan Agency for Medical Research and Development (AMED).

## Author contributions

E.H., S.W. and A.O. conceptualised the study, designed the experiments and wrote the paper. A.O. executed the experiments and analysed the data. A.K. contributed to the adoptive transfer experiments and the drug administration in vivo.

## Additional information

**Competing interests:** The authors declare that they have no competing financial interests.

