## [Peer Review File · Nature Communications]

Reviewers' comments:

Reviewer #1 (Remarks to the Author):

Malignant growth induces the expression of p16INK4a in tumor stromal cells. The authors of this MS tackle the questions of what cells in the tumor stroma express p16INK4a, and delineate the functional effects of such cells. In line with a recent publication by Gudkov and colleagues, the authors show that there is significant expression in monocyte-derived macrophage cells. Surprisingly, the authors show no effect of p16INK4a/p21KIP expression on the cell cycle, and argue that the cells expressing these proteins are not senescent. Instead, the authors argue the expression of p16INK4a/p21KIP promotes the function of myeloid derived suppressor cells in a cell cycle independent manner. These claims are original and interesting, but I have some concerns.

1. I don't think the cell cycle analysis is very convincing. These cells are hyporeplicative, and think the approach taken (DAPI staining) to study their cycling is relatively insensitive and imprecise. It would be more convincing to perform a prolonged in vivo pulse with BrdU (e.g. in the drinking water) and then analyze BrdU incorporation (e.g. over several days) in the MDSC +/- p16INK4a/p21KIP.
2. I am unsure as to whether these cells should be considered senescent. They express p16INK4a and almost certainly would be beta-gal positive at low pH. The absence of DNA damage markers is not especially helpful as many senescent cells do not exhibit sustained DNA damage. Likewise, I don't think the resistance to navitoclax is especially convincing, as other groups have now shown that the effect of this senolytic is cell type specific. Perhaps senescent macrophages are just relatively resistant to navitoclax? I think a careful assessment of the cell cycle status of these cells (see point #1) and expression of SASP transcripts (IL-6, MMP13, etc) would be helpful.
3. I think significant skepticism is warranted regarding the notion that p16INK4a can play an oncogenic role, in this instance by promoting MDSC expression of CX3CR1. While I would be prepared to accept an oncogenic effect for p21KIP, one would be hard pressed to think of any evidence for this with regard to p16INK4a. Keeping in mind that the role of p16INK4a in cancer has been considered in a wide-variety of approaches including human genome-wide surveys, naturally occurring hypomorphic alleles and murine overexpression and knockout studies, the resounding conclusion of these two decades of work has been that p16INK4a represses cancer in most human tissues, and I am unaware of any compelling data to suggest functional p16INK4a can promote tumorigenesis of any tissue. Even if the authors' findings about p16INK4a promoting MDSC function in the context of p21KIP loss are correct, are there data to suggest this would be important in people?
4. I have the most problems in the MS with the authors' proposed model for how p16INK4a/p21KIP promote MDSC function. While I suppose it possible that CDK2 phosphorylates SMAD3, this would mean that p16INK4a/p21KIP were regulating CDK2 activity, which is tightly coupled to the cell cycle, and am surprised the authors are suggesting these CDK inhibitors regulate CDK2 in cells which are not cycling. Also, there are many proteins that control CDK2 activity, many more directly than p16INK4a, and if the authors are correct, expression of any/all of these should affect CX3CR1 expression on MDSC. Again, I think that would amount to the claim that expression of CX3CR1 in MDSC is coupled to the cell cycle. Flavopirodol is a poor choice of reagent to modulate CDK activity in these cells, and it is a very dirty kinase inhibitor with potent activity against many kinases beyond CDK4/2. For these data to be convincing, the authors really need to show that p16INK4a/p21KIP regulate CDK2 activity in non-cycling cells, with corresponding effects on CX3CR1.

Reviewer #2 (Remarks to the Author):

In the present paper, Okuma and Hara provide evidence about the senescence-independent role of p16Ink4a and p21Cip1/waf1 in favoring tumor progression via the accrual of myeloid-derived

suppressor cells (MDSC). By exploiting a well-characterized murine model that allows luciferase-mediated *in vivo* imaging of p16Ink4a and p21Cip1/waf1 expression, the authors report that the progression of a spindle cell tumor (STC) induces the upregulation of these genes in stroma-associated and spleen MDSC, in the absence of any detectable senescent phenotype. The evidence that STC *in vivo* growth is significantly reduced in p16Ink4a and p21Cip1/waf1 knock-out mice clearly indicates a role of these genes in enhancing the protumor effect of MDSC. Through a series of complex and elegant experiments, the authors then demonstrate that the phenomenon occurs through the upregulation of the chemokine receptor CX3CR1, induced by p16Ink4a and p21Cip1/waf1 in MDSC via the prevention of CDK-mediated phosphorylation and SMAD3 inactivation. Finally, Okuma and Hara report that a second tumor cell line (LLC) shows a comparable behavior when transfected with the cognate CX3CR1 ligand, CX3CL1. As underlined by the authors, the evidences that p16Ink4a and p21Cip1/waf1 mediate a protumor effect by regulate the CX3CR1/CX3CL1 axis in MDSC, is a novel and quite interesting finding. Given that therapies aimed at inducing CDK inhibition are an emerging strategy in cancer, the risk of involuntarily accelerating tumor growth by enhancing MDSC accrual represents a serious warning that should be taken into account. The data here provided are technically sound and rather convincing, and results are clearly reported. Nevertheless, some issues should still be clarified and additional experiments could be performed to further strengthen the conclusions, particularly in terms of their potential clinical value.

Major points:

- Figure 1: here the authors report that p16/p21 upregulation in tumor stroma occurs selectively in MDSC. At this regard, it would be useful showing whether other cells of the tumor microenvironment, that could contribute to the protumor effect (such as for instance fibroblasts or endothelial cells), are participating to the process. A multiparametric cytofluorimetry for p16/p21 expression in different subsets of the tumor cell suspension would be perfect. In addition, please indicate when, after tumor injection, the analyses were performed. Finally, check whether reference 15 does address the presence of bone marrow-derived p16Ink4a-expressing MDSC in tumor stroma. Furthermore, I would recommend the authors to discuss some previous reports referring p16Ink4a activation as involved in the induction of M1 polarized myeloid cells (as reviewed in Vicente et al., *Aging Cell* 2016).
- Suppl. Figure 1: here evidences showing no major difference in the MDSC and senescence properties of cells from DKO vs WB mice are reported. MDSC are "fragile" cells *ex vivo*, easily undergoing apoptosis if not supported by specific growth factors; it is thus possible that evaluating cell cycle under resting conditions would not be so informative: in contrast, the authors could test whether upon stimulation (such as for instance IFN γ , or some TLR ligation) MDSC could display senescence traits.
- Suppl figure 2: the data show that no significant differences in the immunosuppressive properties of MDSC from DKO vs WT mice could be observed. However, MDSC might acquire most of their immunosuppressive properties once they reach the tumor site, while their activity is reduced in the periphery (including the spleen, as here utilized) (Marvel and Gabrilovich, *J Clin Invest* 2015). Indeed, the inhibitory activity on T cell proliferation appears to be here around 20-30%. Could the authors test whether tumor-infiltrating MDSC behave differently? To estimate the immunosuppressive properties of tumor-activated MDSC, it would be also useful having some negative controls for comparison, for instance MDSC from non-tumor bearing mice.
- Figure 2: in the experiment on MDSC adoptive transfer, it is hard for me to explain the results of panel e, showing that intratumor MDSC injection does not promote tumor growth in any condition. If the protumor effect of p16/p21 expressing MDSC is due to a more pronounced homing to the tumor, then I would expect that both DKO and WT would have displayed tumor progression. Please, clarify this issue. Please also specify when, with respect to tumor injection, MDSC for *in vivo* transfer were obtained and that WT vs DKO mice bearing the same tumor burden were used, as this could profoundly affect MDSC properties. Furthermore, it would be important corroborating the results of decreased tumor growth in DKO mice by showing immunohistochemical staining of

WT vs DKO, to depict the entity and localization of MDSC infiltrate in the two conditions.

- Suppl. Figure 3. Could the authors please explain if the boost of T cell activation associated with p16/p21 KO was confined to tumor-draining LN or could also be detected at tumor site (as brisk infiltrate)? In addition to CD44 and CD62L, were other T cell activation markers (e.g. PD-1 and other immune checkpoints) modulated?

- Figure 3. The role of the CX3CR1/CX3CL1 axis here depicted is quite convincing. Nevertheless, as other publications have reported the involvement of CCR2/CCL2 or IL-6 in senescent tumor-associated MDSC accrual or activation (Eggert et al., *Cancer Cell* 2016 and Ruhland et al., *Nature Communications* 2016), the expression of these alternative pathways should be also assessed. Data from a second tumor model (LLC) here reported show no change in in vivo growth in DKO vs WT mice, related to the lack of CX3CL1 production by these tumor cells. Are other myeloid chemokines such as CCL2 and CCL5 negative in this cell line as well? To further confirm the role of CX3CL1 in promoting tumor growth of this tumor, it would help knowing whether CX3CL1-transfected LLC cells grow faster than parentale cells in WT mice.

- CX3CR1 and CX3CL1 axis is known to regulate myeloid cell survival by antagonizing pro-apoptotic signals (Landsman et al., *Blood* 2009). Is it possible that an antiapoptotic effect in p16 expressing MDSC could also explain the increased cell accrual? Did the authors tested Bcl2 expression in p16+ MDSC?

- It is unclear whether CX3CL1 is exclusively produced by tumor cells or could be secreted by MDSC as well in a sort of autocrine loop, as it may occur for other chemokines. Did the authors had any chance to test this issue?

- I find rather counterintuitive that p16+ MDSC mediates protumor activity through immunosuppression, but their immunosuppressive properties are not enhanced in WT vs DKO mice. Data here provided support the role of a potentiated MDSC accrual via CX3CR1/CX3CL1, but it would be interesting knowing whether other MDSC protumor functions (e.g proangiogenic effects and EMT) could also be involved. This issue could be addressed by interrogating the RNAseq data provided in figure 3.

- The major limitation of the present work is in my opinion the lack of any parallel in human setting. Some information about the expression rate of CX3CL1 in human cancers, and potential correlation with p16 status, should be searched (even in silico) and discussed. Best option would be to study HPV-related tumors, which are known to express p16 and display a marked immune stroma (Romagosa et al., *Oncogene* 2011; Ferris et al., *J Clin Oncol* 2015).

- Results and discussion are presented in a whole text. I personally believe that using separate paragraphs for the results and a dedicated section for the discussion, would make the reading easier and more incisive. In addition, although the data here provided indicate a senescence-independent protumor role of p16/p21, a possible discussion of the following topics could be considered: senescence-associated secretory profile (SASP), sustaining inflammation and immunosuppression, represents a well-known dark side of senescence (Coppè et al., *Ann Rev Pathol* 2014); p16 expression is associated with MDSC and macrophage accrual in aging human tissues (Ryhland et al., *Nature Commun* 2016); senescence induction in non-tumor cells by chemotherapy promotes toxicity and disease progression (De Maria et al., *Cancer Discovery* 2017); in some models, senescence immune surveillance of premalignant cells prevents cancer development.

Minor:

- Please provide more details about the SCT cell line utilized and the reasons why it was chosen as main model

- Please check whether ref 40 should be deleted and replaced with ref 39

Point-by-point responses to the reviewers' comments

We thank both reviewers for their valuable comments and constructive suggestions. We are pleased that they found our results interesting, and we have tried to address all of the issues that they raised.

Reviewer #1:

Malignant growth induces the expression of p16INK4a in tumor stromal cells. The authors of this MS tackle the questions of what cells in the tumor stroma express p16INK4a, and delineate the functional effects of such cells. In line with a recent publication by Gudkov and colleagues, the authors show that there is significant expression in monocyte-derived macrophage cells. Surprisingly, the authors show no effect of p16INK4a/p21KIP expression on the cell cycle, and argue that the cells expressing these proteins are not senescent. Instead, the authors argue the expression of p16INK4a/p21KIP promotes the function of myeloid derived suppressor cells in a cell cycle independent manner. These claims are original and interesting, but I have some concerns.

1. I don't think the cell cycle analysis is very convincing. These cells are hyporeplicative, and think the approach taken (DAPI staining) to study their cycling is relatively insensitive and imprecise. It would be more convincing to perform a prolonged *in vivo* pulse with BrdU (e.g. in the drinking water) and then analyze BrdU incorporation (e.g. over several days) in the MDSC +/- p16INK4a/p21KIP.

Response-1:

In accordance with the reviewer's suggestion, we have performed the EdU incorporation analysis. Moreover, because the reviewer #2 suggested us to stimulate the cultured MDSCs with growth factors (see Response-5 for reviewer #2), we also performed a cell proliferation analysis *in vitro*. As shown in the revised supplementary fig.1b,c, (methods detail in page 30 line 12 to page 31 line 10), although the proliferation of both PMN-MDSCs and Mo-MDSCs is arrested *in vivo* as judged by EdU incorporation assay regardless of the presence or absence of *p16^{Ink4a}* and *p21^{Cip1/Waf1}* genes, these MDSCs proliferate in response to the GM-CSF stimulation *in vitro*. Thus, it is very unlikely that these MDSCs are in the state of cellular senescence. We have discussed these points in the revised text on page 8 lines 5 to 10.

2. I am unsure as to whether these cells should be considered senescent. They express p16INK4a and almost certainly would be beta-gal positive at low pH. The absence of DNA damage markers is not especially helpful as many senescent cells do not exhibit sustained DNA damage.

Response-2:

“Cellular senescence” is the state of irreversible cell cycle arrest. However, as shown in the revised Supplementary Fig. 1c, significant proportion of the MDSC cells proliferate upon stimulation with GM-CSF *in vitro*. These evidence, together with the absence of other senescence-associated markers (DNA damage response, reduction of lamin B1 expression, or IL-6 expression), indicate that MDSCs are unlikely to be in the state of cellular senescence (revised supplementary Fig. 1d,e,h). It should be noted that SA- β -gal activity is not always associated with cellular senescence (Dimri et al., PNAS, 92, 9363, 1995; Imai et al., Cell Rep. 7, 194, 2014; Krishna et al., Mech Ageing Dev. 109, 113, 1999) and is dispensable for the implementation of cellular senescence (Lee et al., Aging Cell, 5, 187, 2006). Therefore, we are reluctant to use SA- β -gal assay in this experimental setting.

Likewise, I don't think the resistance to navitoclax is especially convincing, as other groups have now shown that the effect of this senolytic is cell type specific. Perhaps senescent macrophages are just relatively resistant to navitoclax?

Response-3:

We agree with this reviewer on this point. However, as described in the Response-1 and -2, the lack of other senescence markers including irreversible cell cycle arrest, strongly suggest that MDSCs are unlikely to be in the state of cellular senescence.

I think a careful assessment of the cell cycle status of these cells (see point #1) and expression of SASP transcripts (IL-6, MMP13, etc) would be helpful.

Response-4:

In line with the reviewer's suggestion, we have examined the levels of IL-6 and MMP13 expression in these MDSCs. As seen in the revised Supplementary Fig. 1g, the levels of IL-6, the most representative SASP factor, in both MDSC subtypes were not higher than those in non-senescent T and B cells. This result also supports our idea that these MDSCs are not in the senescent state. This point is described in revised text on page 8 line 12.

3. I think significant skepticism is warranted regarding the notion that p16INK4a can play an oncogenic role, in this instance by promoting MDSC expression of CX3CR1. While I would be prepared to accept an oncogenic effect for p21KIP, one would be hard pressed to think of any evidence for this with regard to p16INK4a. Keeping in mind that the role of p16INK4a in cancer has been considered in a wide-variety of approaches including human genome-wide surveys, naturally occurring hypomorphic alleles and murine overexpression and knockout studies, the resounding conclusion of these two decades of work has been that p16INK4a represses cancer in most human tissues, and I am unaware of any compelling data to suggest functional p16INK4a can promote tumorigenesis of any tissue. Even if the authors' findings about p16INK4a promoting MDSC function in the context of p21KIP loss are correct, are there data to suggest this would be important in people?

Response-5:

There is no doubt that p16^{Ink4a} is a cell-intrinsic tumour suppressor. However, in the case of p16^{Ink4a} expression in tumour stroma cells, it was previously reported that p16^{Ink4a} expression level was strongly associated with high risk of recurrence or poor survival (Witkiewicz et al., Am J Pathol 179, 1171, 2011; Wang et al., Mol Cancer Res 15, 3, 2017). Certainly, it is difficult to assume a loss of both p16^{Ink4a} and p21^{Cip1/Waf1} in non-tumour cells in patients, but we consider that the possibility of side effects of pharmacologically broad CDK inhibitors is of clinical importance (Fig. 5). We have discussed this point in the revised text on page 20 lines 8 to 13.

4. I have the most problems in the MS with the authors' proposed model for how p16INK4a/p21KIP promote MDSC function. While I suppose it possible that CDK2 phosphorylates SMAD3, this would mean that p16INK4a/p21KIP were regulating CDK2 activity, which is tightly coupled to the cell cycle, and am surprised the authors are suggesting these CDK inhibitors regulate CDK2 in cells which are not cycling. Also, there are many proteins that control CDK2 activity, many more directly than p16INK4a, and if the authors are correct, expression of any/all of these should affect CX3CR1 expression on MDSC. Again, I think that would amount to the claim that expression of CX3CR1 in MDSC is coupled to the cell cycle. Flavopirodol is a poor choice of reagent to modulate CDK activity in these cells, and it is a very dirty kinase inhibitor with potent activity against many kinases beyond CDK4/2. For these data to

be convincing, the authors really need to show that p16INK4a/p21KIP regulate CDK2 activity in non-cycling cells, with corresponding effects on CX3CR1.

Response-6:

Although it was hard to find MDSCs in proliferating phase *in vivo*, we found that these MDSCs are able to proliferate upon stimulation with GM-CSF *in vitro* (revised Supplementary Fig.1a-c, and see also Figure a next page). Notably, the levels of CX3CR1 expression were decreased in proliferating MDSCs (see Figure b next page). Thus, although it is difficult to examine *in vivo*, CX3CR1 expression may be coupled to the cell cycle.

(a) Flow cytometry plots show the relationship between CX3CR1 expression and cell cycle in splenic Mo-MDSCs. DAPI indicates cell cycle state. Total lymphocytes served as a control. There were few Mo-MDSCs in G2/S phase. (b) Flow cytometry plot shows CX3CR1 expression in purified Mo-MDSCs during proliferation. Divided cells were identified as showing diluted CFSE signals relative to unstimulated T cells. CX3CR1 expression were dramatically decreased in dividing cells

We also consider that flavopiridol has broad-spectrum effects (thus, we call flavopiridol a pan-CDK inhibitor). Since we need to inhibit both CDK2 and CDK4 to prevent Smad3 inactivation, we have used flavopiridol. Note that ectopic expression of both CDK2 and CDK4 had a stronger effect to suppress *Cx3cr1* promoter activity as compared with that of

CDK2 or CDK4 alone (see the revised Fig. 4f), further supporting the idea that CDK2 and CDK4 possess the complementary role in inactivating *Cx3cr1* promoter activity. We have discussed these points in the revised text on page 18, lines 12 to 14.

Regarding CDK2 activity in non-cycling MDSCs, we have examined the phosphorylated histone H1, which is a target of CDK2 (J Biol Chem. 268, 1580, 1993; Burstein et al., Mod Pathol, 15, 705, 2002). As shown in figure below, the level of phosphorylated histone H1 was increased in p16/p21-DKO Mo-MDSCs.

Collectively, these data suggest that p16/p21 inhibit CDK2 activity and thereby upregulate *Cx3cr1* expression in non-cycling Mo-MDSCs.

Western blotting of phosphorylated histone H1 was performed with clone 12D11 (Merck; Burstein et al., Mod Pathol, 2002). β -actin served as a loading control. Senescent and asynchronous TIG3, human normal fibroblast cell line, are negative and positive control, respectively.

Reviewer #2:

In the present paper, Okuma and Hara provide evidence about the senescence-independent role of p16Ink4a and p21Cip1/waf1 in favoring tumor progression via the accrual of myeloid-derived suppressor cells (MDSC). By exploiting a well-characterized murine model that allows luciferase-mediated in vivo imaging of p16Ink4a and p21Cip1/waf1 expression, the authors report that the progression of a spindle cell tumor (STC) induces the upregulation of these genes in stroma-associated and spleen MDSC, in the absence of any detectable senescent phenotype. The evidence that STC in vivo growth is significantly reduced in p16Ink4a and p21Cip1/waf1 knock-out mice clearly indicates a role of these genes in enhancing the protumor effect of MDSC. Through a series of complex and elegant experiments, the authors then demonstrate that the phenomenon occurs through the upregulation of the chemokine receptor CX3CR1, induced by p16Ink4a and p21Cip1/waf1 in MDSC via the prevention of CDK-mediated phosphorylation and SMAD3 inactivation. Finally, Okuma and Hara report that a second tumor cell line (LLC) shows a comparable behavior when transfected with the cognate CX3CR1 ligand, CX3CL1.

As underlined by the authors, the evidences that p16Ink4a and p21Cip1/waf1 mediate a protumor effect by regulate the CX3CR1/CX3CL1 axis in MDSC, is a novel and quite interesting finding. Given that therapies aimed at inducing CDK inhibition are an emerging strategy in cancer, the risk of involuntarily accelerating tumor growth by enhancing MDSC accrual represents a serious warning that should be taken into account.

The data here provided are technically sound and rather convincing, and results are clearly reported. Nevertheless, some issues should still be clarified and additional experiments could be performed to further strenghten the conclusions, particularly in terms of their potential clinical value.

Major points:

- Figure 1: here the authors report that p16/p21 upregulation in tumor stroma occurs selectively in MDSC. At this regard, it would be useful showing whether other cells of the tumor microenvironment, that could contribute to the protumor effect (such as for instance fibroblasts or endothelial cells), are participating to the process. A multiparametric cytofluorimetry for p16/p21 expression in different subsets of the tumor cell suspension would be perfect.

Response-1:

We agree with the reviewer that it would be great if we could perform the experiments suggested by the reviewer. We have tried to test many commercially available antibodies

against p16^{Ink4a} and p21^{Cip1/Waf1}, and found that none of them showed sufficient quality other than M156 (anti-p16; Santa Cruz) and ab2961 (anti-p21; Abcam). However, these antibodies are not commercially available anymore. Thus, unfortunately, it is difficult for us to perform the experiments suggested by the reviewer.

In addition, please indicate when, after tumor injection, the analyses were performed.

Response-2:

We apologize for omission of this condition. We have added the information in the revised figure legend (see the revised text on page 45, lines 3, 4, 7, 8, 12, and 14).

Finally, check whether reference 15 does address the presence of bone marrow-derived p16Ink4a-expressing MDSC in tumor stroma.

Response-3:

We deeply apologize for our mistake. We have rearranged the reference list.

Furthermore, I would recommend the authors to discuss some previous reports referring p16Ink4a activation as involved in the induction of M1 polarized myeloid cells (as reviewed in Vicente et al., Aging Cell 2016).

Response-4:

We think this is an important suggestion. Accordingly, we have included new discussion about this point (new text on page 20 line 14 to page 21 line 5). Note that we also focused on M1/M2 polarization initially. However, the levels of p16 expression in tumour-associated macrophages (F4/80+ cells) are lower than those in MDSCs in this experimental setting (see figure below). Furthermore, p16/p21-DKO macrophages should be M2-prone. Indeed, our data also show that Arg1, which is a M2 marker, is up-regulated in DKO Mo-MDSCs (Supplementary Fig.2d). It is believed that M2 polarization represents a pro-tumour/anti-inflammatory state in macrophages; therefore, we consider that M1/M2 polarization is incompatible with this p16/p21-mediated pro-tumour phenotype.

qPCR assay for detection of p16. TAMs were isolated from SCT-bearing WT mice, using magnetic beads (Anti-F4/80 MicroBeads UltraPure, mouse; Miltenyi Biotech).

- Suppl. Figure 1: here evidences showing no major difference in the MDSC and senescence properties of cells from DKO vs WB mice are reported. MDSC are “fragile” cells ex vivo, easily undergoing apoptosis if not supported by specific growth factors; it is thus possible that evaluating cell cycle under resting conditions would not be so informative: in contrast, the authors could test whether upon stimulation (such as for instance IFN γ , or some TLR ligation) MDSC could display senescence traits.

Response-5:

Please see the Response-1 and -2 for reviewer #1. Our data show that MDSCs could proliferate upon GM-CSF stimulation (see the revised Supplementary Fig. 1b and c). Additionally, we also showed that IL-6 and IFN- γ , which promote MDSC differentiation *in vitro*, accelerate MDSC proliferation (see the figure below). These evidence, together with the absence of other senescence-associated markers (DNA damage response, reduction of Lamin B1 expression, or IL-6 expression), indicate that MDSCs are unlikely to be in the state of cellular senescence (revised supplementary Fig. 1d-g).

Purified Mo-MDSCs and PMN-MDSCs were cultured in indicated cytokine-containing medium for 3 days. Histograms indicate CFSE signals in MDSCs. Divided cells were identified as showing diluted CFSE signals relative to unstimulated T cells. Method details are presented in new text, page 29 line 6 to page 30 line 4.

- Suppl figure 2: the data show that no significant differences in the immunosuppressive properties of MDSC from DKO vs WT mice could be observed. However, MDSC might acquire most of their immunosuppressive properties once they reach the tumor site, while their activity is reduced in the periphery (including the spleen, as here utilized) (Marvel and Gabrilovich, J Clin Invest 2015). Indeed, the inhibitory activity on T cell proliferation appears to be here around 20-30%. Could the authors test whether tumor-infiltrating MDSC behave differently? To estimate the immunosuppressive properties of tumor-activated MDSC, it would be also useful having some negative controls for comparison, for instance MDSC from non-tumor bearing mice.

Response-6:

It should be noted that both PMN-MDSCs and Mo-MDSCs from the spleen sufficiently inhibited antigen-specific T-cell responses (>70% inhibition; Supplementary Fig.2a). In the case of antigen-independent T-cell proliferation, the inhibitory activity of MDSCs is certainly low even when considering intratumour MDSCs (Supplementary Fig.2b and new Supplementary Fig.2c). MDSCs are rarely detectable in non-tumour-bearing mice; therefore, we used BM-derived dendritic cells as a negative control in some cases, as shown in new Supplementary Fig.7b (GM).

- Figure 2: in the experiment on MDSC adoptive transfer, it is hard for me to explain the results of panel e, showing that intratumour MDSC injection does not promote tumor growth in any condition. If the protumor effect of p16/p21 expressing MDSC is due to a more pronounced homing to the tumor, then I would expect that both DKO and WT would have displayed tumor progression. Please, clarify this issue. Please also specify when, with respect to tumor injection, MDSC for in vivo transfer were obtained and that WT vs DKO mice bearing the same tumor burden were used, as this could profoundly affect MDSC properties. Furthermore, it would be important corroborating the results of decreased tumor growth in DKO mice by showing immunohistochemical staining of WT vs DKO, to depict the entity and localization of MDSC infiltrate in the two conditions.

Response-7:

We deeply apologize for having caused this confusion. In Fig. 2e, intratumour Mo-MDSC injection promoted tumour growth regardless of the presence or absence of *p16^{Ink4a}* and *p21^{Cip1/Waf1}* expression in MDSCs. We have included statistical results in new Fig. 2e. With regard to the suggested IHC, we confirmed intratumour infiltration of transferred Mo-MDSCs (new Fig.2f and revised text on page 11 line 15 to page 12 line 2). To distinguish between

transferred Mo-MDSCs and host Mo-MDSCs, purified Mo-MDSCs were stained with CFSE before transfer. Apparent difference in intratumour infiltration was observed between WT Mo-MDSCs and DKO Mo-MDSCs in the case of intravenously injection but not intratumour injection.

- Suppl. Figure 3. Could the authors please explain if the boost of T cell activation associated with p16/p21 KO was confined to tumor-draining LN or could also be detected at tumor site (as brisk infiltrate)? In addition to CD44 and CD62L, were other T cell activation markers (e.g. PD-1 and other immune checkpoints) modulated?

Response-8:

We think this is a very important suggestion. Accordingly, we have evaluated the T cell activation status in tumours (new Supplementary Fig.3e-h and figures below). Most intratumour CD8 T cells were in the CD44^{high} population, but the expression level of CD69 was increased in p16/p21-DKO mice. These data suggest that the difference in T cell activation between WT and p16/p21-DKO mice occurs not only in draining LN but also at the tumour site. With regard to PD-1, we believe that currently, PD-1 is considered an exhaustion marker rather than an activation marker. It is unclear whether the change of PD-1 expression is due to the effect of MDSCs, but we consider that intratumour CD8 T cells in p16/p21-DKO mice are kept in an activated state. We have described these points in the revised text on page 12 lines 9 to 14.

Representative flow cytometry plots of the CD3+CD8+ population in tumours. Graph shows the mean of the CD44high population (n = 4).

- **Figure 3. The role of the CX3CR1/CX3CL1 axis here depicted is quite convincing. Nevertheless, as other publications have reported the involvement of CCR2/CCL2 or IL-6 in senescent tumor-associated MDSC accrual or activation (Eggert et al., Cancer Cell 2016 and Ruhland et al., Nature Communications 2016), the expression of these alternative pathways should be also assessed. Data from a second tumor model (LLC) here reported show no change in in vivo growth in DKO vs WT mice, related to the lack of CX3CL1 production by these tumor cells. Are other myeloid chemokines such as CCL2 and CCL5 negative in this cell line as well?**

Response-9:

This series of suggestions is very important, and we have also approached these issues.

We assessed CCR2 and CCR5 function as well, because these were also identified as chemokine receptors whose expression appears to be correlated with the presence of p16/p21, as judged by RNA sequencing analysis. As shown, decline of CCR5 expression in p16/p21-DKO Mo-MDSCs was not confirmed by flow cytometry (revised Supplementary Fig.4d,e). Expression of CCR2 ligands, CCL2 and CCL7, in SCT were lower than in LLC, which grows normally even in p16/p21-DKO mice, although SCT expresses more CX3CL1 than LLC (revised Supplementary Fig.5a-d). Therefore, we concluded that the change in CX3CR1 expression is most important for SCT growth due to the consistency of the results. We have described these points in the revised text on page 13 lines 9 to 10, and page 14 line 11 to 13.

Although LLC express high CCL2 and CCL7, intratumour Mo-MDSCs in LLC are less than in SCT (Figure below left). We speculate that low expression of GM-CSF, an inducer of MDSCs, is one of the cause of poor Mo-MDSC recruitment (Figure below right).

Left graph shows population of Mo-MDSCs and PMN-MDSCs in CD45+ cells in SCT and LLC. Tumours were resected from WT mice at a size of 500 mm³. Right graph shows that *Csf2* (GM-SCF) expression in LLC and SCT. Data are presented as the mean \pm SEM (n = 4).

Regarding that tumour-associated senescent cells recruit MDSCs to tumour, CAF and pre-malignant cells were not observed in our experimental setting. On the other hand, we speculate that ageing might accelerate tumour progression due to accumulation of senescent cells and up-regulation of p16/p21 in MDSCs. We discussed this point in revised text on page 22 lines 9 to 16.

To further confirm the role of CX3CL1 in promoting tumor growth of this tumor, it would help knowing whether CX3CL1-transfected LLC cells grow faster than parentale cells in WT mice.

Response-10:

As shown in the revised Fig.3f and Supplementary Fig.5a, CX3CL1-transfected LLCs grow obviously slower than control LLCs. These data are consistent with those of a previously reported study, in which authors showed that CX3CL1 chemoattracts anti-tumour T lymphocytes, resulting in decreased tumour growth (Guo et al., Int J Cancer. 103, 212-20, 2003). In our case, we expect that the difference in growth of CX3CL1-expressing LLCs between WT and DKO is caused not only by amplified MDSC attraction but also by activated tumour immunity, because MDSCs inhibit tumour immunity, causing tumour growth to be accelerated.

- CX3CR1 and CX3CL1 axis is known to regulate myeloid cell survival by antagonizing pro-apoptotic signals (Landsman et al., Blood 2009). Is it possible that an antiapoptotic effect in p16 expressing MDSC could also explain the increased cell accrual? Did the authors tested Bcl2 expression in p16+ MDSC? Response-11:

We did not know about the paper reported by Landsman et al.; We thank the reviewer for this information. As shown in the revised supplementary Fig.6, we were unable to detect substantial difference in the expression of *Bcl2* and apoptosis between WT and p16/p21-DKO MDSCs. These results are described in the revised text on page 15, lines 9 to 15.

- It is unclear whether CX3CL1 is exclusively produced by tumor cells or could be secreted by MDSC as well in a sort of autocrine loop, as it may occur for other chemokines. Did the authors had any chance to test this issue?

Response-12:

As shown new Supplementary Fig. 5b, there was no difference of CX3CL1 expression level between SCT cell lines cultured *in vitro* and SCTs inoculated *in vivo*, including tumour stromal cells. Furthermore, CX3CL1 expression in MDSCs was much less than in SCT (see Figure below).

Cx3cl1 mRNA expression level in indicated cell types was measured by qPCR assay. Graphs

indicates means +SEM.

- I find rather counterintuitive that p16+ MDSC mediates protumor activity through immunosuppression, but their immunosuppressive properties are not enhanced in WT vs DKO mice. Data here provided support the role of a potentiated MDSC accrual via CX3CR1/CX3CL1, but it would be interesting knowing whether other MDSC protumor functions (e.g proangiogenic effects and EMT) could also be involved. This issue could be addressed by interrogating the RNAseq data provided in figure 3.

Response-13:

We think this is an important suggestion. Accordingly, we have analysed the difference of proangiogenic factors and EMT-related genes between WT Mo-MDSCs and p16/p21-DKO Mo-MDSCs. RNA-sequencing data show that no factors that regulate angiogenesis or EMT were decreased in p16/p21-DKO Mo-MDSCs (see new Supplementary Table 1 & 2 and text on page 10 line 15 to page 11 line 3 in the revised manuscript).

- The major limitation of the present work is in my opinion the lack of any parallel in human setting. Some information about the expression rate of CX3CL1 in human cancers, and potential correlation with p16 status, should be searched (even in silico) and discussed. Best option would be to study HPV-related tumors, which are known to express p16 and display a marked immune stroma (Romagosa et al., Oncogene 2011; Ferris et al., J Clin Oncol 2015)

Response-14:

We appreciate this suggestion. Regarding CX3CL1 expression in human cancer, we have added a discussion in the text on page 21 line 6 to page 22 line 3 in the revised manuscript. However, it should be noted that we did not mention the expression of p16^{Ink4a} in cancer cells. Indeed, some types of cancer, including HPV-related tumours, express p16^{Ink4a} at high levels, but it is thought that the cause of p16^{Ink4a} overexpression is inactivation of Rb. Rb (retinoblastoma protein) is a downstream factor of the p16^{Ink4a} pathway and a key regulator of G1 cell cycle arrest. In our study, we consider that p16^{Ink4a} expression in the stroma is important. We have included a relevant discussion in the text on page 20 line 10 to 12 in the revised manuscript.

- Results and discussion are presented in a whole text. I personally believe that using separate paragraphs for the results and a dedicated section for the discussion, would make the reading easier and more incisive.

Response-15:

We apologize for the format of this manuscript. We have revised it as the reviewer suggested.

In addition, although the data here provided indicate a senescence-independent protumor role of p16/p21, a possible discussion of the following topics could be considered: senescence-associated secretory profile (SASP), sustaining inflammation and immunosuppression, represents a well-known dark side of senescence (Coppè et al., Ann Rev Pathol 2014); p16 expression is associated with MDSC and macrophage accrual in aging human tissues (Ryhland et al., Nature Commun 2016); senescence induction in non-tumor cells by chemotherapy promotes toxicity and disease progression (De Maria et al., Cancer Discovery 2017); in some models, senescence immune surveillance of premalignant cells prevents cancer development.

Response-18:

As the reviewer mentioned, SASP is certainly a dark side of cellular senescence. However, it was previously reported that p16^{Ink4a} and p21^{Cip1/Waf1} do not regulate SASP factor expression (Coppé et al., J Biol Chem. 286, 36396, 2011). Indeed, the expression levels of various SASP factors were not decreased in p16/p21 DKO Mo-MDSCs compared with WT Mo-MDSCs (new Supplementary Fig.1g and RNA-seq data, table below).

Gene	DKO/WT Fold Change	P-Value	sesssed log2 (Centering)						WT FPKM	DKO FPKM
			WT			DKO				
Il1a	1.120	0.559	0	0	0	0	0	0	10.889	12.360
Il1b	-1.316	0.210	0	0	0	0	0	0	33.054	25.269
Il6	1.151	0.422	0	0	0	0	0	0	0.035	0.085
Cxcl1	1.000	NaN	0	0	0	0	0	0	0.000	0.000
Ccl2	3.283	0.024	0	0	0	0	0	0	0.821	2.725
Mmp13	2.387	0.006	0	0	0	0	0	0	1.497	3.567
Csf2	1.000	NaN	0	0	0	0	0	0	0.000	0.028
Serpine1	1.540	0.053	0	0	0	0	0	0	1.714	2.616
Igfbp1	1.000	NaN	0	0	0	0	0	0	0.039	0.019
Igfbp4	-1.337	0.272	0	0	0	0	0	0	5.147	3.746

Table showed expression change of representative SASP factors between WT and p16/p21DKO Mo-MDSCs

Regarding the association of p27^{Kip1}-induced senescent tissues with MDSCs, the paper mentioned by the reviewer is interesting. In their system, p27^{Kip1}-induced cells were restricted to fibroblasts because the Cre driver is the *Colla2* promoter. MDSC accumulation appears to be induced by increased expression of chemokines such as CCL2. We speculate that p16^{Ink4a} and p21^{Cip1/Waf1} in MDSCs are also involved in this senescent cell-induced inflammatory phenomenon. We discussed this point in revised text on page 22 lines 9 to 16 and see also Response-9.

Regarding therapy-induced senescence, it should be noted that MDSCs are highly sensitive to most anti-cancer drugs that kill proliferating cells (Wesolowski et al., J Immunother Cancer. 1, 10, 2013). In most cases, therapy-induced senescent cells are premalignant cancer cells or tumour-associated fibroblasts, and MDSCs are transiently eliminated.

Regarding immune surveillance of senescence, we speculate that MDSCs help senescent cells to survive; therefore, p16^{Ink4a} and p21^{Cip1/Waf1} in MDSCs may promote not only tumour progression but also cancer development.

Minor:

- Please provide more details about the SCT cell line utilized and the reasons why it was chosen as main model

Response-19:

SCT (spindle cell tumour) is a malignant skin cancer line and derived from DMBA plus TPA-treated p16-KO mouse (Takeuchi et al., Cancer Res, 70, 9381, 2010). Tumour subcutaneous inoculation is a representative easy technique. In this case of using SCT, we consider that this experimental setting is an orthotopic implantation and relatively natural. We have included this point in revised text on page 9 lines 8 and 9, page 24 line 12.

- Please check whether ref 40 should be deleted and replaced with ref 39

Response-20:

We deeply apologize for our mistake. We have rearranged the reference list.

Reviewers' comments:

Reviewer #1 (Remarks to the Author):

I don't find this version more convincing than the original submission. The main issue for me remains the mechanism whereby p16/p21 are supposed to influence MDSC function. The authors on the one hand claim that MDSC from WT and DKO mice have similar proliferative rates and suggest these effects of p16/p21 are E2F independent and "novel" (which would suggest they mean cell cycle independent, since an effect of these proteins on the cell cycle is well known). On the other hand, the authors suggest these proteins are producing these "novel" effects on motility, chemokine receptor expression etc via an effect of these on CDK2/4 activity (which they claim phosphorylate SMAD3). Leaving aside the criticism that the data to support SMAD3 as CDK4 substrate are pretty weak, this model just does not make a lot of sense: it suggests the activity of CDK2/4 is different in non-cycling MDSCs in a manner depending on p16/p21 but not to a degree to influence the cell cycle? Especially when one considers how much CDK2 activity varies btw G1 and S phase, this model seems very implausible.

I also have to repeat my concerns about the use of flavopirodol for the experiments described. This kinase inhibitor is incredibly dirty, with profound effects on CDK1/5/7/9 etc as well as non-CDK kinases. It has broad effects on global transcription, and its use to approximate a CDK4/CDK2 inhibitor in vivo is sloppy.

Reviewer #2 (Remarks to the Author):

I am grateful to the authors for having so seriously and effectively addressed all the issues raised by my revision. I have no additional concern, and I believe that the manuscript is now presenting convincing evidence that p16Ink4a and p21Cip1/Waf1 may promote tumor growth by acting on the activity of myeloid-derived suppressor cells. Given the increasing role that these cells are gaining in human cancer, particularly as ultimate frontier to immune-mediated cancer control that we now know occurring with most cancer therapies, the data here reported offer a new and key view on the involvement of these cells in tumor progression and how to intervene for their in vivo blunting.

Point-by-point responses to the reviewers' comments

We would like to thank reviewers for their valuable comments and important suggestions. Since remained concerns arise from our insufficient explanations, we have conducted additional experiments to provide much clearer mechanistic information and tried to address all concerns adequately.

Reviewer #1 (Remarks to the Author):

Point-1:

I don't find this version more convincing than the original submission. The main issue for me remains the mechanism whereby p16/p21 are supposed to influence MDSC function. The authors on the one hand claim that MDSC from WT and DKO mice have similar proliferative rates and suggest these effects of p16/p21 are E2F independent and "novel" (which would suggest they mean cell cycle independent, since an effect of these proteins on the cell cycle is well known). On the other hand, the authors suggest these proteins are producing these "novel" effects on motility, chemokine receptor expression etc via an effect of these on CDK2/4 activity (which they claim phosphorylate SMAD3). Leaving aside the criticism that the data to support SMAD3 as CDK4 substrate are pretty weak, this model just does not make a lot of sense: it suggests the activity of CDK2/4 is different in non-cycling MDSCs in a manner depending on p16/p21 but not to a degree to influence the cell cycle? Especially when one considers how much CDK2 activity varies btw G1 and S phase, this model seems very implausible.

Response:

We deeply apologize for our insufficient explanation based only on non-proliferative MDSCs. In order to better explain, we have compared the effects of p16/p21 on cell

cycle profile, CDK2 activity, phosphorylation status of SMAD3 and CX3CR1 expression in Mo-MDSCs between bone marrow (proliferating) Mo-MDSCs and splenic (non-proliferating) Mo-MDSCs.

Consistent with a previous report (Nat. Immunol. 14, 211-220, 2013), a significant proportion of bone marrow Mo-MDSCs are proliferating (see new Supplementary Fig. 1a, upper left graph). However, once Mo-MDSCs leave bone marrow, they start to mature and slow down their cell cycle, as seen in splenic Mo-MDSCs (see new Supplementary Fig. 1a, upper middle graph). This non-dividing state is largely independent of p16/p21, because the cell cycle profiles are basically the same regardless of p16/p21 gene status. We have described these points in the re-revised text on page 8 lines 2 to 7.

Because p16^{Ink4a} and p21^{Cip1/Waf1} cooperatively suppress CDK2 activity (Mol. Cell. Biol. 19, 1981-1989, 1999), we have measured CDK2 kinase activity in Mo-MDSCs purified from bone marrow or spleen using SMAD3 as a substrate. As shown in new Fig. 4d, although the level of CDK2 activity in splenic Mo-MDSCs was lower than that in bone marrow Mo-MDSCs, a certain level of CDK2 activity was still existing in splenic Mo-MDSCs. Note that these levels were higher in p16/p21-DKO mice as compared to those in WT mice, indicating that p16^{Ink4a} and p21^{Cip1/Waf1} CDK inhibitors are functioning even in non-proliferating splenic Mo-MDSCs. Importantly, moreover, the levels of CDK2 activity in these Mo-MDSCs were positively or negatively correlated with those of phosphorylated SMAD3 or CX3CR1 expression, respectively (see new Figs. 3c, 4d,e and schematic diagram below), suggesting that p16/p21 play important roles in controlling CX3CR1 expression in both bone marrow Mo-MDSCs and splenic Mo-MDSCs. These results are somewhat consistent with recent observations that CDK2/4 also plays cell-cycle independent roles in various non-proliferating cells, such as in neuronal cells (see review in Nat. Rev. Mol. Cell Biol., 17, 280-292, 2016). We have described these points in the re-revised text on page 18 lines 9 to 13, page 19 lines 8 to 12, and page 23 lines 6 to page 24 line 1.

We agree that it is important to include above points in our manuscript. Thus, we are truly grateful to the reviewer # 1 for this important suggestion.

Point-2:

I also have to repeat my concerns about the use of flavopirodol for the experiments described. This kinase inhibitor is incredibly dirty, with profound effects on CDK1/5/7/9 etc as well as non-CDK kinases. It has broad effects on global transcription, and its use to approximate a CDK4/CDK2 inhibitor in vivo is sloppy.

Response:

Since the specificity of the chemical CDK inhibitors are certainly important for analyzing their activity, we obtained several kinds of reagents, LY2835219 (CDK4/6-specific inhibitor; Invest New Drugs, 32, 825-837, 2014), PD 0332991 (CDK4/6-specific inhibitor; Mol Cancer Ther., 3, 1427-1438, 2004), NU6027 (CDK1/2-specific inhibitor; J Med Chem., 43, 2797-804, 2000), and K03861 (CDK2-specific inhibitor; ACS Chem Biol., 10, 2116-25, 2015), that could inhibit CDK activity more specifically than flavopiridol and examined their effect on the expression of *Cx3cr1* mRNA in Mo-MDSCs. In addition to flavopiridol, CDK2 inhibitors, NU6027 and K03861, induced/de-repressed

Cx3cr1 expression, whereas CDK4/6 inhibitors, LY2835219 and PD 0332991, had little effect on it (see new Fig. 4c). We have described these points in the re-revised text on page 18 lines 6 to 13.

Besides the fact that p16/p21 cooperatively suppresses CDK2 activity, our data indicate that there was not statistically significant difference in the tumour progression in p16 single KO mice or p21 single KO mice compared to that in wildtype mice (see the figure below). Together, it is most likely that CDK2 inhibitor rather than CDK4 inhibitor could induce *Cx3cr1* expression.

Tumour size in SCT-injected male WT, p16-KO, p21-KO, and p16/p21-DKO mice. NS, not significant; ***, $P < 0.001$

Reviewer #2 (Remarks to the Author):

I am grateful to the authors for having so seriously and effectively addressed all the issues raised by my revision. I have no additional concern, and I believe that the manuscript is now presenting convincing evidence that p16^{Ink4a} and p21^{Cip1/Waf1} may promote tumor growth by acting on the activity of myeloid-derived suppressor cells. Giving the increasing role that these cells are gaining in human cancer, particularly as ultimate frontier to immune-mediated cancer control that we now know occurring with most cancer therapies, the data here

reported offer a new and key view on the involvement of these cells in tumor progression and hoiw to intervene for their in vivo blunting.

Response:

Thank you very much.

REVIEWERS' COMMENTS:

Reviewer #3 (Remarks to the Author):

In this second revision, the authors have addressed the previous reviewers remaining concerns. They provide a reasonable explanation for the role of CDK2 (and maybe CDK4) on Smad3 activity in non-proliferative cells. It may, however, be beneficial to stress that the observed effect on Smad3 could be indirect. The authors have also addressed issues of CDK inhibitor specificity and toxicity, using a range of additional drugs. While genetic manipulation would be preferable, I understand this is difficult in the cell types in question.

I have no further concerns and recommend publication of this manuscript.